# A Neural Tangent Kernel Perspective of Infinite Tree Ensembles

**Ryuichi Kanoh[1,2], Mahito Sugiyama[1,2]**
[1]National Institute of Informatics
[2]The Graduate University for Advanced Studies, SOKENDAI
{kanoh, mahito}@nii.ac.jp

## Abstract

In practical situations, the tree ensemble is one of the most popular models along with neural networks. A *soft tree* is a variant of a decision tree. Instead of using a greedy method for searching splitting rules, the soft tree is trained using a gradient method in which the entire splitting operation is formulated in a differentiable form. Although ensembles of such soft trees have been used increasingly in recent years, little theoretical work has been done to understand their behavior. By considering an ensemble of *infinite* soft trees, this paper introduces and studies the *Tree Neural Tangent Kernel* (TNTK), which provides new insights into the behavior of the infinite ensemble of soft trees. Using the TNTK, we theoretically identify several non-trivial properties, such as global convergence of the training, the equivalence of the oblivious tree structure, and the degeneracy of the TNTK induced by the deepening of the trees.

## 1 Introduction

Tree ensembles and neural networks are powerful machine learning models that are used in various real-world applications. A *soft tree ensemble* is one variant of tree ensemble models that inherits characteristics of neural networks. Instead of using a greedy method (Quinlan, 1986; Breiman et al., 1984) to search splitting rules, the soft tree makes the splitting rules *soft* and updates the entire model's parameters simultaneously using the gradient method. Soft tree ensemble models are known to have high empirical performance (Kontschieder et al., 2015; Popov et al., 2020; Hazimeh et al., 2020), especially for tabular datasets. Apart from accuracy, there are many reasons why one should formulate trees in a soft manner. For example, unlike hard decision trees, soft tree models can be updated sequentially (Ke et al., 2019) and trained in combination with pre-training (Arik & Pfister, 2019), resulting in characteristics that are favorable in terms of real-world continuous service deployment. Their model interpretability, induced by the hierarchical splitting structure, has also attracted much attention (Frosst & Hinton, 2017; Wan et al., 2021; Tanno et al., 2019). In addition, the idea of the soft tree is implicitly used in many different places; for example, the process of allocating data to the appropriate leaves can be interpreted as a special case of Mixture-of-Experts (Jordan & Jacobs, 1993; Shazeer et al., 2017; Lepikhin et al., 2021), a technique for balancing computational complexity and prediction performance.

Although various techniques have been proposed to train trees, the theoretical validity of such techniques is not well understood at sufficient depth. Examples of the practical technique include constraints on individual trees using parameter sharing (Popov et al., 2020), adjusting the hardness of the splitting operation (Frosst & Hinton, 2017; Hazimeh et al., 2020), and the use of overparameterization (Belkin et al., 2019; Karthikeyan et al., 2021). To better understand the training of tree ensemble models, we focus on the *Neural Tangent Kernel* (NTK) (Jacot et al., 2018), a powerful tool that has been successfully applied to various neural network models with *infinite* hidden layer nodes. Every model architecture is known to produce a distinct NTK. Not only for the multi-layer perceptron (MLP), many studies have been conducted across various models, such as for Convolutional Neural Networks (CNTK) (Arora et al., 2019; Li et al., 2019), Graph Neural Networks (GNTK) (Du et al., 2019b), and Recurrent Neural Networks (RNTK) (Alemohammad et al., 2021). Although a number of findings have been obtained using the NTK, they are mainly for typical neural networks, and it is still not clear how to apply the NTK theory to the tree models.

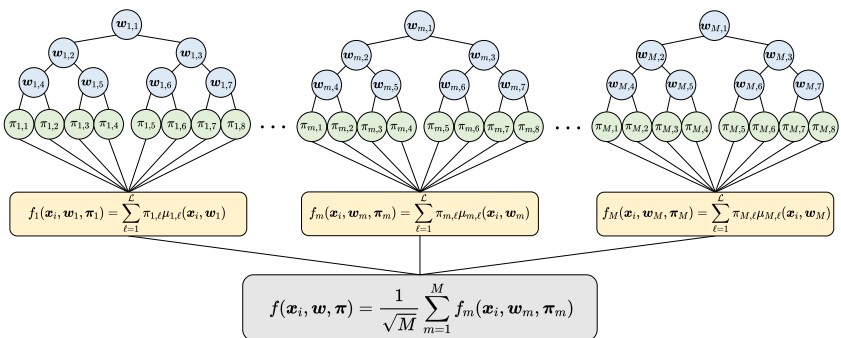

Figure 1: Schematics of an ensemble of $M$ soft trees. Tree internal nodes are indexed according to the breadth-first ordering.

In this paper, by considering the limit of infinitely many trees, we introduce and study the neural tangent kernel for tree ensembles, called the *Tree Neural Tangent Kernel* (TNTK), which provides new insights into the behavior of the ensemble of soft trees. The goal of this research is to derive the kernel that characterizes the training behavior of soft tree ensembles, and to obtain theoretical support for the empirical techniques. Our contributions are summarized as follows:

- **First extension of the NTK concept to the tree ensemble models.** We derive the analytical form for the TNTK at initialization induced by infinitely many perfect binary trees with arbitrary depth (Section 4.1.1). We also prove that the TNTK remains constant during the training of infinite soft trees, which allows us to analyze the behavior by kernel regression and discuss global convergence of training using the positive definiteness of the TNTK (Section 4.1.2, 4.1.3).

- **Equivalence of the oblivious tree ensemble models.** We show the TNTK induced by the oblivious tree structure used in practical open-source libraries such as CatBoost (Prokhorenkova et al., 2018) and NODE (Popov et al., 2020) converges to the same TNTK induced by a non-oblivious one in the limit of infinite trees. This observation implicitly supports the good empirical performance of oblivious trees with parameter sharing between tree nodes (Section 4.2.1).

- **Nonlinearity by adjusting the tree splitting operation.** Practically, various functions have been proposed to represent the tree splitting operation. The most basic function is sigmoid. We show that the TNTK is almost a linear kernel in the basic case, and when we adjust the splitting function hard, the TNTK becomes nonlinear (Section 4.2.2).

- **Degeneracy of the TNTK with deep trees.** The TNTK associated with deep trees exhibits degeneracy: the TNTK values are almost identical for deep trees even if the inner products of inputs are different. As a result, poor performance in numerical experiments is observed with the TNTK induced by infinitely many deep trees. This result supports the fact that the depth of trees is usually not so large in practical situations (Section 4.2.3).

- **Comparison to the NTK induced by the MLP.** We investigate the generalization performance of infinite tree ensembles by kernel regression with the TNTK on 90 real-world datasets. Although the MLP with infinite width has better prediction accuracy on average, the infinite tree ensemble performs better than the infinite width MLP in more than 30 percent of the datasets. We also showed that the TNTK is superior to the MLP-induced NTK in computational speed (Section 5).

## 2 BACKGROUND AND RELATED WORK

Our main focus in this paper is the soft tree and the neural tangent kernel. We briefly introduce and review them.

### 2.1 SOFT TREE

Based on Kontschieder et al. (2015), we formulate a regression by soft trees. Figure 1 is a schematic image of an ensemble of $M$ soft trees. We define a data matrix $\boldsymbol{x} \in \mathbb{R}^{F \times N}$ for $N$ training samples

$\{\boldsymbol{x}_1, \ldots, \boldsymbol{x}_N\}$ with $F$ features and define tree-wise parameter matrices for internal nodes $\boldsymbol{w}_m \in \mathbb{R}^{F \times \mathcal{N}}$ and leaf nodes $\boldsymbol{\pi}_m \in \mathbb{R}^{1 \times \mathcal{L}}$ for each tree $m \in [M] = \{1, \ldots, M\}$ as

$$
\boldsymbol{x} = \left( \begin{array}{ccc} | & \cdots & | \\ \boldsymbol{x}_1 & \cdots & \boldsymbol{x}_N \\ | & \cdots & | \end{array} \right), \quad
\boldsymbol{w}_m = \left( \begin{array}{ccc} | & \cdots & | \\ \boldsymbol{w}_{m,1} & \cdots & \boldsymbol{w}_{m,\mathcal{N}} \\ | & \cdots & | \end{array} \right), \quad
\boldsymbol{\pi}_m = (\pi_{m,1}, \ldots, \pi_{m,\mathcal{L}}),
$$

where internal nodes (blue nodes in Figure 1) and leaf nodes (green nodes in Figure 1) are indexed from 1 to $\mathcal{N}$ and 1 to $\mathcal{L}$, respectively. $\mathcal{N}$ and $\mathcal{L}$ may change across trees in general, while we assume that they are always fixed for simplicity throughout the paper. We also write horizontal concatenation of (column) vectors as $\boldsymbol{x} = (\boldsymbol{x}_1, \ldots, \boldsymbol{x}_N) \in \mathbb{R}^{F \times N}$ and $\boldsymbol{w}_m = (\boldsymbol{w}_{m,1}, \ldots, \boldsymbol{w}_{m,\mathcal{N}}) \in \mathbb{R}^{F \times \mathcal{N}}$. Unlike hard decision trees, we consider a model in which every single leaf node $\ell \in [\mathcal{L}] = \{1, \ldots, \mathcal{L}\}$ of a tree $m$ holds the probability that data will reach to it. Therefore, the splitting operation at an intermediate node $n \in [\mathcal{N}] = \{1, \ldots, \mathcal{N}\}$ does not definitively decide splitting to the left or right. To provide an explicit form of the probabilistic tree splitting operation, we introduce the following binary relations that depend on the tree's structure: $\ell \swarrow n$ (resp. $n \searrow \ell$), which is true if a leaf $\ell$ belongs to the left (resp. right) subtree of a node $n$ and false otherwise. We can now exploit $\mu_{m,\ell}(\boldsymbol{x}_i, \boldsymbol{w}_m) : \mathbb{R}^F \times \mathbb{R}^{F \times \mathcal{N}} \to [0, 1]$, a function that returns the probability that a sample $\boldsymbol{x}_i$ will reach a leaf $\ell$ of the tree $m$, as follows:

$$
\mu_{m,\ell}(\boldsymbol{x}_i, \boldsymbol{w}_m) = \prod_{n=1}^{\mathcal{N}} g_{m,n}(\boldsymbol{x}_i, \boldsymbol{w}_{m,n})^{\mathbb{1}_{\ell \swarrow n}} \left( 1 - g_{m,n}(\boldsymbol{x}_i, \boldsymbol{w}_{m,n}) \right)^{\mathbb{1}_{n \searrow \ell}}, \tag{1}
$$

where $\mathbb{1}_Q$ is an indicator function conditioned on the argument $Q$, i.e., $\mathbb{1}_{\text{true}} = 1$ and $\mathbb{1}_{\text{false}} = 0$, and $g_{m,n} : \mathbb{R}^F \times \mathbb{R}^F \to [0, 1]$ is a decision function at each internal node $n$ of a tree $m$. To approximate decision tree splitting, the output of the decision function $g_{m,n}$ should be between 0.0 and 1.0. If the output of a decision function takes only 0.0 or 1.0, the splitting operation is equivalent to hard splitting used in typical decision trees. We will define an explicit form of the decision function $g_{m,n}$ in Equation (5) in the next section.

The prediction for each $\boldsymbol{x}_i$ from a tree $m$ with nodes parameterized by $\boldsymbol{w}_m$ and $\boldsymbol{\pi}_m$ is given by

$$
f_m(\boldsymbol{x}_i, \boldsymbol{w}_m, \boldsymbol{\pi}_m) = \sum_{\ell=1}^{\mathcal{L}} \pi_{m,\ell} \mu_{m,\ell}(\boldsymbol{x}_i, \boldsymbol{w}_m), \tag{2}
$$

where $f_m : \mathbb{R}^F \times \mathbb{R}^{F \times \mathcal{N}} \times \mathbb{R}^{1 \times \mathcal{L}} \to \mathbb{R}$, and $\pi_{m,\ell}$ denotes the response of a leaf $\ell$ of the tree $m$. This formulation means that the prediction output is the average of the leaf values $\pi_{m,\ell}$ weighted by $\mu_{m,\ell}(\boldsymbol{x}_i, \boldsymbol{w}_m)$, probability of assigning the sample $\boldsymbol{x}_i$ to the leaf $\ell$. If $\mu_{m,\ell}(\boldsymbol{x}_i, \boldsymbol{w}_m)$ takes only 1.0 for one leaf and 0.0 for the other leaves, the behavior is equivalent to a typical decision tree prediction. In this model, $\boldsymbol{w}_m$ and $\boldsymbol{\pi}_m$ are updated during training with a gradient method.

While many empirical successes have been reported, theoretical analysis for soft tree ensemble models has not been sufficiently developed.

## 2.2 NEURAL TANGENT KERNEL

Given $N$ samples $\boldsymbol{x} \in \mathbb{R}^{F \times N}$, the NTK induced by any model architecture at a training time $\tau$ is formulated as a matrix $\widehat{\boldsymbol{H}}_\tau^* \in \mathbb{R}^{N \times N}$, in which each $(i, j) \in [N] \times [N]$ component is defined as

$$
[\widehat{\boldsymbol{H}}_\tau^*]_{ij} := \widehat{\Theta}_\tau^*(\boldsymbol{x}_i, \boldsymbol{x}_j) := \left\langle \frac{\partial f_{\text{arbitrary}}(\boldsymbol{x}_i, \boldsymbol{\theta}_\tau)}{\partial \boldsymbol{\theta}_\tau}, \frac{\partial f_{\text{arbitrary}}(\boldsymbol{x}_j, \boldsymbol{\theta}_\tau)}{\partial \boldsymbol{\theta}_\tau} \right\rangle, \tag{3}
$$

where $\langle \cdot, \cdot \rangle$ denotes the inner product and $\boldsymbol{\theta}_\tau \in \mathbb{R}^P$ is a concatenated vector of all the $P$ trainable model parameters at $\tau$. An asterisk " * " indicates that the model is arbitrary. The model function $f_{\text{arbitrary}} : \mathbb{R}^F \times \mathbb{R}^P \to \mathbb{R}$ used in Equation (3) is expected to be applicable to a variety of model structures. For the soft tree ensembles introduced in Section 2.1, the NTK is formulated as $\sum_{m=1}^{M} \sum_{n=1}^{\mathcal{N}} \left\langle \frac{\partial f(\boldsymbol{x}_i, \boldsymbol{w}, \boldsymbol{\pi})}{\partial \boldsymbol{w}_{m,n}}, \frac{\partial f(\boldsymbol{x}_j, \boldsymbol{w}, \boldsymbol{\pi})}{\partial \boldsymbol{w}_{m,n}} \right\rangle + \sum_{m=1}^{M} \sum_{\ell=1}^{\mathcal{L}} \left\langle \frac{\partial f(\boldsymbol{x}_i, \boldsymbol{w}, \boldsymbol{\pi})}{\partial \pi_{m,\ell}}, \frac{\partial f(\boldsymbol{x}_j, \boldsymbol{w}, \boldsymbol{\pi})}{\partial \pi_{m,\ell}} \right\rangle$.

Within the limit of infinite width with a proper parameter scaling, a variety of properties have been discovered from the NTK induced by the MLP. For example, Jacot et al. (2018) showed the convergence of $\widehat{\Theta}_0^{\text{MLP}}(\boldsymbol{x}_i, \boldsymbol{x}_j)$, which can vary with respect to parameters, to the unique limiting

kernel $\Theta^{\mathrm{MLP}}(\boldsymbol{x}_i, \boldsymbol{x}_j)$ at initialization in probability. Moreover, they also showed that the limiting kernel does not change during training in probability:

$$\lim_{\text{width}\to\infty} \widehat{\Theta}_\tau^{\mathrm{MLP}}(\boldsymbol{x}_i, \boldsymbol{x}_j) = \lim_{\text{width}\to\infty} \widehat{\Theta}_0^{\mathrm{MLP}}(\boldsymbol{x}_i, \boldsymbol{x}_j) = \Theta^{\mathrm{MLP}}(\boldsymbol{x}_i, \boldsymbol{x}_j). \tag{4}$$

This property helps in the analytical understanding of the model behavior. For example, with the squared loss and infinitesimal step size with learning rate $\eta$, the training dynamics of gradient flow in function space coincides with kernel ridge-less regression with the limiting NTK. Such a property gives us a data-dependent generalization bound (Bartlett & Mendelson, 2003) related to the NTK and the prediction targets. In addition, if the NTK is positive definite, the training can achieve global convergence (Du et al., 2019a; Jacot et al., 2018).

Although a number of findings have been obtained using the NTK, they are mainly for typical neural networks such as MLP and ResNet (He et al., 2016), and the NTK theory has not yet been applied to tree models. The NTK theory is often used in the context of overparameterization, which is a subject of interest not only for the neural networks, but also for the tree models (Belkin et al., 2019; Karthikeyan et al., 2021; Tang et al., 2018).

## 3 SETUP

We train model parameters $\boldsymbol{w}$ and $\boldsymbol{\pi}$ to minimize the squared loss using the gradient method, where $\boldsymbol{w} = (\boldsymbol{w}_1, \ldots, \boldsymbol{w}_M)$ and $\boldsymbol{\pi} = (\boldsymbol{\pi}_1, \ldots, \boldsymbol{\pi}_M)$. The tree structure is fixed during training. In order to use a known closed-form solution of the NTK (Williams, 1996; Lee et al., 2019), we use a scaled error function $\sigma : \mathbb{R} \to (0, 1)$, resulting in the following decision function:

$$g_{m,n}(\boldsymbol{x}_i, \boldsymbol{w}_{m,n}) = \sigma\left(\boldsymbol{w}_{m,n}^\top \boldsymbol{x}_i\right)$$
$$:= \frac{1}{2}\,\mathrm{erf}\left(\alpha \boldsymbol{w}_{m,n}^\top \boldsymbol{x}_i\right) + \frac{1}{2}, \tag{5}$$

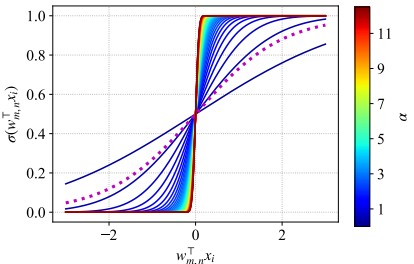

Figure 2: The scaled error function. We draw 50 lines with varying $\alpha$ by 0.25. A dotted magenta line shows a sigmoid function, which is close to a scaled error function with $\alpha \sim 0.5$.

where $\mathrm{erf}(p) = \frac{2}{\sqrt{\pi}} \int_0^p e^{-t^2}\, \mathrm{d}t$ for $p \in \mathbb{R}$. This scaled error function approximates a commonly used sigmoid function. Since the bias term for the input of $\sigma$ can be expressed inside of $\boldsymbol{w}$ by adding an element that takes a fixed constant value for all input of the soft trees $\boldsymbol{x}$, we do not consider the bias for simplicity. The scaling factor $\alpha$ is introduced by Frosst & Hinton (2017) to avoid overly soft splitting. Figure 2 shows that the decision function becomes harder as $\alpha$ increases (from blue to red), and in the limit $\alpha \to \infty$ it coincides with the hard splitting used in typical decision trees.

When aggregating the output of multiple trees, we divide the sum of the tree outputs by the square root of the number of trees

$$f(\boldsymbol{x}_i, \boldsymbol{w}, \boldsymbol{\pi}) = \frac{1}{\sqrt{M}} \sum_{m=1}^{M} f_m(\boldsymbol{x}_i, \boldsymbol{w}_m, \boldsymbol{\pi}_m). \tag{6}$$

This $1/\sqrt{M}$ scaling is known to be essential in the existing NTK literature to use the weak law of the large numbers (Jacot et al., 2018). On top of Equation (6), we initialize each of model parameters $\boldsymbol{w}_{m,n}$ and $\pi_{m,\ell}$ with zero-mean i.i.d. Gaussians with unit variances. We refer such a parameterization as *NTK initialization*. In this paper, we consider a model such that all $M$ trees have the same perfect binary tree structure, a common setting for soft tree ensembles (Popov et al., 2020; Kontschieder et al., 2015; Hazimeh et al., 2020).

## 4 THEORETICAL RESULTS

### 4.1 BASIC PROPERTIES OF THE TNTK

The NTK in Equation (3) induced by the soft tree ensembles is referred to here as the TNTK and denoted by $\widehat{\Theta}_0^{(d)}(\boldsymbol{x}_i, \boldsymbol{x}_j)$ the TNTK at initialization induced by the ensemble of trees with depth $d$.

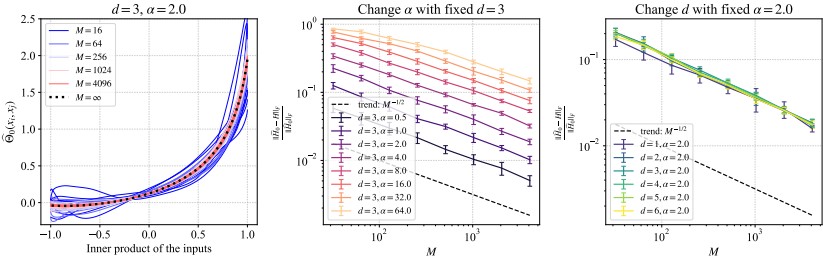

Figure 3: Left: An empirical demonstration of convergence of $\widehat{\Theta}_0(\boldsymbol{x}_i, \boldsymbol{x}_j)$ to the fixed limit $\Theta(\boldsymbol{x}_i, \boldsymbol{x}_j)$ as $M$ increases. Two simple inputs are considered: $\boldsymbol{x}_i = \{1, 0\}$ and $\boldsymbol{x}_j = \{\cos(\beta), \sin(\beta)\}$ with $\beta = [0, \pi]$. The TNTK $\widehat{\Theta}_0^{(3)}(\boldsymbol{x}_i, \boldsymbol{x}_j)$ with $\alpha = 2.0$ is calculated 10 times with parameter re-initialization for each of the $M = 16, 64, 256, 1024$, and $4096$. Center and Right: Parameter dependency of the convergence. The vertical axis corresponds to the averaged error between the $\widehat{\boldsymbol{H}}_0$ and the $\boldsymbol{H} := \lim_{M \to \infty} \widehat{\boldsymbol{H}}_0$ for 50 random unit vectors of length $F = 5$. The dashed lines are plotted only for showing the slope. The error bars show the standard deviations of 10 executions.

In this section, we show the properties of the TNTK that are important for understanding the training behavior of the soft tree ensembles.

### 4.1.1 TNTK FOR INFINITE TREE ENSEMBLES

First, we show the formula of the TNTK at initialization, which converges when considering the limit of infinite trees ($M \to \infty$).

**Theorem 1.** *Let $\boldsymbol{u} \in \mathbb{R}^F$ be any column vector sampled from zero-mean i.i.d. Gaussians with unit variance. The TNTK for an ensemble of soft perfect binary trees with tree depth $d$ converges in probability to the following deterministic kernel as $M \to \infty$,*

$$
\begin{aligned}
\Theta^{(d)}(\boldsymbol{x}_i, \boldsymbol{x}_j) &:= \lim_{M \to \infty} \widehat{\Theta}_0^{(d)}(\boldsymbol{x}_i, \boldsymbol{x}_j) \\
&= \underbrace{2^d d\, \Sigma(\boldsymbol{x}_i, \boldsymbol{x}_j)(\mathcal{T}(\boldsymbol{x}_i, \boldsymbol{x}_j))^{d-1}\dot{\mathcal{T}}(\boldsymbol{x}_i, \boldsymbol{x}_j)}_{\text{contribution from inner nodes}} + \underbrace{(2\mathcal{T}(\boldsymbol{x}_i, \boldsymbol{x}_j))^d}_{\text{contribution from leaves}},
\end{aligned}
\tag{7}
$$

*where $\Sigma(\boldsymbol{x}_i, \boldsymbol{x}_j) := \boldsymbol{x}_i^\top \boldsymbol{x}_j$, $\mathcal{T}(\boldsymbol{x}_i, \boldsymbol{x}_j) := \mathbb{E}[\sigma(\boldsymbol{u}^\top \boldsymbol{x}_i)\sigma(\boldsymbol{u}^\top \boldsymbol{x}_j)]$, and $\dot{\mathcal{T}}(\boldsymbol{x}_i, \boldsymbol{x}_j) := \mathbb{E}[\dot{\sigma}(\boldsymbol{u}^\top \boldsymbol{x}_i)\dot{\sigma}(\boldsymbol{u}^\top \boldsymbol{x}_j)]$. Moreover, $\mathcal{T}(\boldsymbol{x}_i, \boldsymbol{x}_j)$ and $\dot{\mathcal{T}}(\boldsymbol{x}_i, \boldsymbol{x}_j)$ are analytically obtained in the closed-form as*

$$
\mathcal{T}(\boldsymbol{x}_i, \boldsymbol{x}_j) = \frac{1}{2\pi}\arcsin\left(\frac{\alpha^2 \Sigma(\boldsymbol{x}_i, \boldsymbol{x}_j)}{\sqrt{(\alpha^2\Sigma(\boldsymbol{x}_i, \boldsymbol{x}_i) + 0.5)(\alpha^2\Sigma(\boldsymbol{x}_j, \boldsymbol{x}_j) + 0.5)}}\right) + \frac{1}{4},
\tag{8}
$$

$$
\dot{\mathcal{T}}(\boldsymbol{x}_i, \boldsymbol{x}_j) = \frac{\alpha^2}{\pi}\frac{1}{\sqrt{(1 + 2\alpha^2\Sigma(\boldsymbol{x}_i, \boldsymbol{x}_i))(1 + 2\alpha^2\Sigma(\boldsymbol{x}_j, \boldsymbol{x}_j)) - 4\alpha^4\Sigma(\boldsymbol{x}_i, \boldsymbol{x}_j)^2}}.
\tag{9}
$$

The dot used in $\dot{\sigma}(\boldsymbol{u}^\top \boldsymbol{x}_i)$ means the first derivative: $\alpha e^{-(\alpha \boldsymbol{u}^\top \boldsymbol{x}_i)^2}/\sqrt{\pi}$, and $\mathbb{E}[\cdot]$ means the expectation. The scalar $\pi$ in Equation (8) and Equation (9) is the circular constant, and $\boldsymbol{u}$ corresponds to $\boldsymbol{w}_{m,n}$ at an arbitrary internal node. The proof is given by induction. We can derive the formula of the limiting TNTK by treating the number of trees in a tree ensemble like the width of the hidden layer in MLP, although the MLP and the soft tree ensemble are apparently different models. Due to space limitations, detailed proofs are given in the supplementary material.

We demonstrate convergence of the TNTK in Figure 3. We empirically observe that the TNTK induced by sufficiently many soft trees converges to the limiting TNTK given in Equation (7). The kernel values induced by an finite ensemble are already close to the limiting TNTK if the number of trees is larger than several hundreds, which is a typical order of the number of trees in practical applications[1]. Therefore, it is reasonable to analyze soft tree ensembles via the TNTK.

---

[1] For example, Popov et al. (2020) uses 2048 trees.

By comparing Equation (7) and the limiting NTK induced by a two-layer perceptron (shown in the supplementary material), we can immediately derive the following when the tree depth is 1.

**Corollary 1.** *If the splitting function at the tree internal node is the same as the activation function of the neural network, the limiting TNTK obtained from a soft tree ensemble of depth 1 is equivalent to the limiting NTK generated by a two-layer perceptron up to constant multiple.*

For any tree depth larger than 1, the limiting NTK induced by the MLP with any number of layers (Arora et al., 2019, shown in the supplementary material) and the limiting TNTK do not match. This implies that the hierarchical splitting structure is a distinctive feature of soft tree ensembles.

### 4.1.2 POSITIVE DEFINITENESS OF THE LIMITING TNTK

Since the loss surface of a large model is expected to be highly non-convex, understanding the good empirical trainability of overparameterized models remains an open problem (Dauphin et al., 2014). The positive definiteness of the limiting kernel is one of the most important conditions for achieving global convergence (Du et al., 2019a; Jacot et al., 2018). Jacot et al. (2018) showed that the conditions $\|\boldsymbol{x}_i\|_2 = 1$ for all $i \in [N]$ and $\boldsymbol{x}_i \neq \boldsymbol{x}_j$ $(i \neq j)$ are necessary for the positive definiteness of the NTK induced by the MLP for an input set. As for the TNTK, since the formulation (Equation (7)) is different from that of typical neural networks such as an MLP, it is not clear whether or not the limiting TNTK is positive definite.

We prove that the TNTK induced by infinite trees is also positive definite under the same condition for the MLP.

**Proposition 1.** *For infinitely many soft trees with any depth and the NTK initialization, the limiting TNTK is positive definite if $\|\boldsymbol{x}_i\|_2 = 1$ for all $i \in [N]$ and $\boldsymbol{x}_i \neq \boldsymbol{x}_j$ $(i \neq j)$.*

The proof is provided in the supplementary material. Similar to the discussion for the MLP (Du et al., 2019a; Jacot et al., 2018), if the limiting TNTK is constant during training, the positive definiteness of the limiting TNTK at initialization indicates that training of the infinite trees with a gradient method can converge to the global minimum. The constantness of the limiting TNTK during training is shown in the following section.

### 4.1.3 CHANGE OF THE TNTK DURING TRAINING

We prove that the TNTK hardly changes from its initial value during training when considering an ensemble of infinite trees with finite $\alpha$ (used in Equation (5)).

**Theorem 2.** *Let $\lambda_{\min}$ and $\lambda_{\max}$ be the minimum and maximum eigenvalues of the limiting TNTK. Assume that the limiting TNTK is positive definite for input sets. For soft tree ensemble models with the NTK initialization and a positive finite scaling factor $\alpha$ trained under gradient flow with a learning rate $\eta < 2/(\lambda_{\min} + \lambda_{\max})$, we have, with high probability,*

$$\sup \left| \widehat{\Theta}_\tau^{(d)}(\boldsymbol{x}_i, \boldsymbol{x}_j) - \widehat{\Theta}_0^{(d)}(\boldsymbol{x}_i, \boldsymbol{x}_j) \right| = \mathcal{O}\left( \frac{1}{\sqrt{M}} \right). \tag{10}$$

The complete proof is provided in the supplementary material. Figure 4 shows that the training trajectory analytically obtained (Jacot et al., 2018; Lee et al., 2019) from the limiting TNTK and the trajectory during gradient descent training become similar as the number of trees increases, demonstrating the validity of using the TNTK framework to analyze the training behavior.

**Remarks.** In the limit of infinitely large $\alpha$, which corresponds to a hard decision tree splitting (Figure 2), it should be noted that this theorem does not hold because of the lack of local Lipschitzness (Lee et al., 2019), which is the fundamental property for this proof. Therefore the change in the TNTK during training is no longer necessarily asymptotic to zero, even if the number of trees is infinite. This means that understanding the hard decision tree's behavior using the TNTK is not straightforward.

### 4.2 IMPLICATIONS FOR PRACTICAL TECHNIQUES

In this section, from the viewpoint of the TNTK, we discuss the training techniques that have been used in practice.

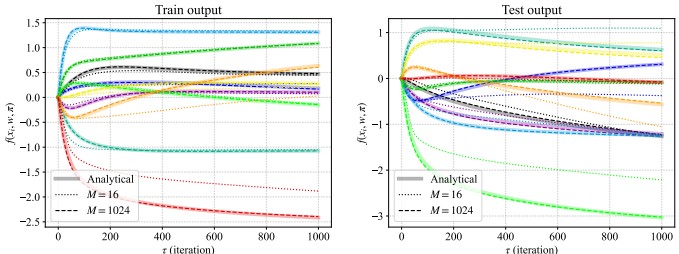

Figure 4: Output dynamics for train and test data points. The color of each line corresponds to each data point. Soft tree ensembles with $d = 3$, $\alpha = 2.0$ are trained by a full-batch gradient descent with a learning rate of $0.1$. Initial outputs are shifted to zero (Chizat et al., 2019). There are $10$ randomly generated training points and $10$ randomly generated test data points, and their dimension $F = 5$. The prediction targets are also randomly generated. Let $\boldsymbol{H}(\boldsymbol{x}, \boldsymbol{x}') \in \mathbb{R}^{N \times N'}$ be the limiting NTK matrix for two input matrices and $\boldsymbol{I}$ be an identity matrix. For analytical results, we draw the trajectory $f(\boldsymbol{v}, \boldsymbol{\theta}_\tau) = \boldsymbol{H}(\boldsymbol{v}, \boldsymbol{x})\boldsymbol{H}(\boldsymbol{x}, \boldsymbol{x})^{-1}(\boldsymbol{I} - \exp[-\eta \boldsymbol{H}(\boldsymbol{x}, \boldsymbol{x})\tau])\boldsymbol{y}$ (Lee et al., 2019) using the limiting TNTK (Equation (7)), where $\boldsymbol{v} \in \mathbb{R}^F$ is an arbitrary input and $\boldsymbol{x} \in \mathbb{R}^{F \times N}$ and $\boldsymbol{y} \in \mathbb{R}^N$ are the training dataset and the targets, respectively.

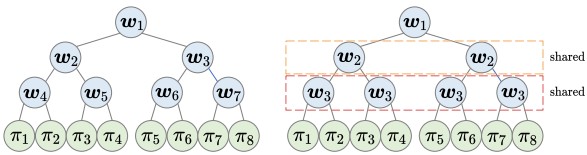

Figure 5: Left: Normal Tree, Right: Oblivious Tree. The rules for splitting in the same depth are shared across the same depth in the oblivious tree, while $\pi_{m,\ell}$ on leaves can be different.

### 4.2.1 INFLUENCE OF THE OBLIVIOUS TREE STRUCTURE

An *oblivious tree* is a practical tree model architecture where the rules across decision tree splitting are shared across the same depth as illustrated in Figure 5. Since the number of the splitting decision calculation can be reduced from $\mathcal{O}(2^d)$ to $\mathcal{O}(d)$, the oblivious tree structure is used in various open-source libraries such as CatBoost (Prokhorenkova et al., 2018) and NODE (Popov et al., 2020). However, the reason for the good empirical performance of oblivious trees is non-trivial despite weakening the expressive power due to parameter sharing.

We find that the oblivious tree structure does not change the limiting TNTK from the non-oblivious one. This happens because, even with parameter sharing at splitting nodes, leaf parameters $\boldsymbol{\pi}$ are not shared, resulting in independence between outputs of left and right subtrees.

**Theorem 3.** *The TNTK with the perfect binary tree ensemble and the TNTK of its corresponding oblivious tree ensemble obtained via parameter sharing converge to the same kernel in probability in the limit of infinite trees $(M \to \infty)$.*

The complete proof is in the supplementary material. Note that leaf values $\boldsymbol{\pi}$ do not have to be the same for oblivious and non-oblivious trees. This theorem supports the recent success of tree ensemble models with the oblivious tree structures.

### 4.2.2 EFFECT OF THE DECISION FUNCTION MODIFICATION

Practically, based on the commonly used $\mathrm{sigmoid}$ function, a variety of functions have been proposed for the splitting operation. By considering a large scaling factor $\alpha$ in Equation (5), we can envisage situations in which there are practically used hard functions, such as two-class $\mathrm{sparsemax}$, $\sigma(x) = \mathrm{sparsemax}([x, 0])$ (Martins & Astudillo, 2016), and two-class $\mathrm{entmax}$, $\sigma(x) = \mathrm{entmax}([x, 0])$ (Peters et al., 2019). Figure 6 shows $\alpha$ dependencies of TNTK parameters. Equation (7) means that the TNTK is formulated by multiplying $\mathcal{T}(\boldsymbol{x}_i, \boldsymbol{x}_j)$ and $\dot{\mathcal{T}}(\boldsymbol{x}_i, \boldsymbol{x}_j)$ to the linear kernel $\Sigma(\boldsymbol{x}_i, \boldsymbol{x}_j)$. On the one hand, $\mathcal{T}(\boldsymbol{x}_i, \boldsymbol{x}_j)$ and $\dot{\mathcal{T}}(\boldsymbol{x}_i, \boldsymbol{x}_j)$ with small $\alpha$ are almost con-

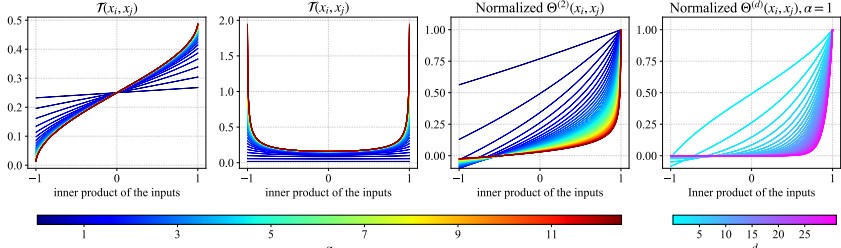

Figure 6: Parameter dependencies of $\mathcal{T}(\boldsymbol{x}_i, \boldsymbol{x}_j)$, $\dot{\mathcal{T}}(\boldsymbol{x}_i, \boldsymbol{x}_j)$, and $\Theta^{(d)}(\boldsymbol{x}_i, \boldsymbol{x}_j)$. The vertical axes are normalized so that the value is $1$ when the inner product of the inputs is $1$. The input vector size is normalized to be one. For the three figures on the left, the line color is determined by $\alpha$, and for the figure on the right, it is determined by the depth of the tree.

stant, even for different input inner products, resulting in the almost linear TNTK. On the other hand, the nonlinearity increases as $\alpha$ increases. For Figure 2, the original sigmoid function corresponds to a scaled error function for $\alpha \sim 0.5$, which induces almost the linear kernel. Although a closed-form TNTK using sigmoid as a decision function has not been obtained, its kernel is expected to be almost linear. Therefore, from the viewpoint of the TNTK, an adjustment of the decision function (Frosst & Hinton, 2017; Popov et al., 2020; Hazimeh et al., 2020) can be interpreted as an escape from the linear kernel behavior.

### 4.2.3 DEGENERACY CAUSED BY DEEP TREES

As the depth increases, $\mathcal{T}(\boldsymbol{x}_i, \boldsymbol{x}_j)$ defined in Equation (8), which consists of the arcsine function, is multiplied multiple times to calculate the limiting TNTK in Equation (7). Therefore, when we increase the depth too much, the resulting TNTK exhibits degeneracy: its output values are almost the same as each other's, even though the input's inner products are different. The rightmost panel of Figure 6 shows such degeneracy behavior. In terms of kernel regression, models using a kernel that gives almost the same inner product to all data except those that are quite close to each other are expected to have poor generalization performance. Such behavior is observed in our numerical experiments (Section 5). In practical applications, overly deep soft or hard decision trees are not usually used because overly deep trees show poor performance (Luo et al., 2021), which is supported by the degeneracy of the TNTK.

## 5 NUMERICAL EXPERIMENTS

**Setup.** We present our experimental results on $90$ classification tasks in the UCI database (Dua & Graff, 2017), with fewer than $5000$ data points, as in Arora et al. (2020). We performed kernel regression using the limiting TNTK defined in Equation (7) with varying the tree depth ($d$) and the scaling ($\alpha$) of the decision function. The limiting TNTK does not change during training for an infinite ensemble of soft trees (Theorem 2); therefore, predictions from that model are equivalent to kernel regression using the limiting TNTK (Jacot et al., 2018). To consider the ridge-less situation, regularization strength is set to be $1.0 \times 10^{-8}$, a very small constant. By way of comparison, performances of the kernel regression with the MLP-induced NTK (Jacot et al., 2018) and the RBF kernel are also reported. For the MLP-induced NTK, we use ReLU for the activation function. We follow the procedures of Arora et al. (2020) and Fernández-Delgado et al. (2014): We report $4$-fold cross-validation performance with random data splitting. To tune parameters, all available training samples are randomly split into one training and one validation set, while imposing that each class has the same number of training and validation samples. Then the parameter with the best validation accuracy is selected. Other details are provided in the supplementary material.

**Comparison to the MLP.** The left panel of Figure 7 shows the averaged performance as a function of the depth. Although the TNTK with properly tuned parameters tend to be better than those obtained with the RBF kernel, they are often inferior to the MLP-induced NTK. The results support the good performance of the MLP-induced NTK (Arora et al., 2020). However, it should be noted that when we look at each dataset one by one, the TNTK is superior to the MLP-induced NTK by more than

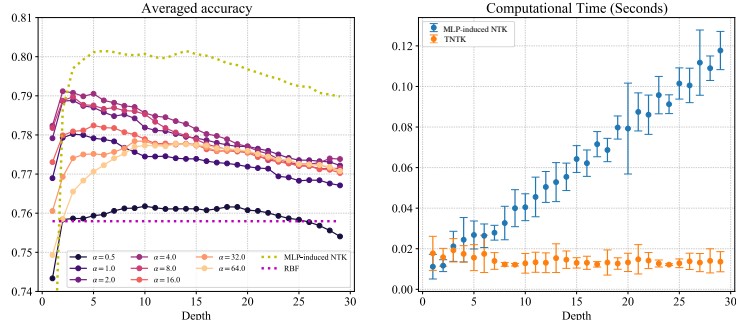

Figure 7: Left: Averaged accuracy over 90 datasets. The performances of the kernel regression with the MLP-induced NTK and the RBF kernel are shown for comparison. Since the depth is not a hyper-parameter of the RBF kernel, performance is shown by a horizontal line. The statistical significance is also assessed in the supplementary material. Right: Running time for kernel computation. The input dataset has 300 samples with 10 features. Feature values are generated by zero-mean i.i.d Gaussian with unit variance. The error bars show the standard deviations of 10 executions.

Table 1: Performance win rate against the MLP-induced NTK. We tune the depth from $d = 1$ to 29 for the dataset-wise comparison for both the TNTK and the MLP-induced NTK. For the RBF kernel, 30 different hyperparameters are tried. Detailed results are in the supplementary material.

|  | TNTK | | | | | | | | RBF |
|---|---|---|---|---|---|---|---|---|---|
| $\alpha$ | 0.5 | 1.0 | 2.0 | 4.0 | 8.0 | 16.0 | 32.0 | 64.0 | — |
| Win rate (%) | 13.6 | 18.8 | 22.2 | 28.6 | 32.5 | 31.6 | 34.9 | 27.2 | 11.8 |

30 percent of the dataset, as shown in Table 1. This is a case where the characteristics of data and the inductive bias of the model fit well together. In addition, although the computational cost of the MLP-induced NTK is linear with respect to the depth of the model because of the recursive computation (Jacot et al., 2018), the computational cost of the TNTK does not depend on the depth, as shown in Equation (7). This results in much faster computation than the MLP-induced NTK when the depth increases, as illustrated in the right panel of Figure 7. Even if the MLP-induced NTK is better in prediction accuracy, the TNTK may be used in practical cases as a trade-off for computational complexity. Arora et al. (2019) proposed the use of the NTK for a neural architecture search (Elsken et al., 2019; Chen et al., 2021). In such applications, the fast computation of the kernel in various architectures can be a benefit. We leave extensions of this idea to tree models as future work.

**Consistency with implications from the TNTK theory.** When we increase the tree depth, we initially observe an improvement in performance, after which the performance gradually decreases. This behavior is consistent with the performance deterioration due to degeneracy (Section 4.2.3), similar to that reported for neural networks without skip-connection (Huang et al., 2020), shown by a dotted yellow line. The performance improvement by adjusting $\alpha$ in the decision function (Frosst & Hinton, 2017) is also observed. Performances with hard ($\alpha > 0.5$) decision functions are always better than the sigmoid-like function ($\alpha = 0.5$, as shown in Figure 2).

## 6 CONCLUSION

In this paper, we have introduced and studied the *Tree Neural Tangent Kernel* (TNTK) by considering the ensemble of infinitely many soft trees. The TNTK provides new insights into the behavior of the infinite ensemble of soft trees, such as the effect of the oblivious tree structure and the degeneracy of the TNTK induced by the deepening of the trees. In numerical experiments, we have observed the degeneracy phenomena induced by the deepening of the soft tree model, which is suggested by our theoretical results. To date, the NTK theory has been mostly applied to neural networks, and our study is the first to apply it to the tree model. Therefore our study represents a milestone in the development of the NTK theory.

ACKNOWLEDGEMENT

This work was supported by JSPS KAKENHI (Grant Number JP21H03503d, Japan), JST PRESTO (Grant Number JPMJPR1855, Japan), and JST FOREST (Grant Number JPMJFR206J, Japan).

ETHICS STATEMENT

We believe that the TNTK's theoretical analysis will not have a negative impact on society as our work does not directly lead to harmful applications.

REPRODUCIBILITY STATEMENT

Proofs of all theoretical results are provided in the supplementary material. For the numerical experiments and figures, we share our reproducible source code in the supplementary material.

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

## A    PROOF OF THEOREM 1

*Proof.* The model output from a certain depth tree ensemble $f^{(d)}$ can be written alternatively using an incremental formula as

$$f^{(d)}(\boldsymbol{x}_i, \boldsymbol{w}, \boldsymbol{\pi}) = \frac{1}{\sqrt{M}} \sum_{m=1}^{M} \left( \sigma\left(\boldsymbol{w}_{m,t}^{\top} \boldsymbol{x}_i\right) f_m^{(d-1)}\left(\boldsymbol{x}_i, \boldsymbol{w}_m^{(l)}, \boldsymbol{\pi}_m^{(l)}\right) \right.$$
$$\left. + \left(1 - \sigma\left(\boldsymbol{w}_{m,t}^{\top} \boldsymbol{x}_i\right)\right) f_m^{(d-1)}\left(\boldsymbol{x}_i, \boldsymbol{w}_m^{(r)}, \boldsymbol{\pi}_m^{(r)}\right) \right), \qquad \text{(A.1)}$$

where indices $(l)$ and $(r)$ used with model parameters $\boldsymbol{w}_m$ and $\boldsymbol{\pi}_m$ mean the parameters at the (l)eft subtree and the (r)ight subtree, respectively, and $t$ used in $\boldsymbol{w}_{m,t}$ denotes the node at (t)op of the tree. For example, for trees of depth 3, as shown in Figure 1, $\boldsymbol{w}_m^{(l)} = (\boldsymbol{w}_{m,2}, \boldsymbol{w}_{m,4}, \boldsymbol{w}_{m,5})$, $\boldsymbol{w}_m^{(r)} = (\boldsymbol{w}_{m,3}, \boldsymbol{w}_{m,6}, \boldsymbol{w}_{m,7})$, and $\boldsymbol{w}_{m,t} = \boldsymbol{w}_{m,1}$.

We prove the theorem by induction. When $d = 1$,

$$f^{(1)}(\boldsymbol{x}_i, \boldsymbol{w}, \boldsymbol{\pi}) = \frac{1}{\sqrt{M}} \sum_{m=1}^{M} \left( \sigma(\boldsymbol{w}_{m,1}^{\top} \boldsymbol{x}_i) \pi_{m,1} + (1 - \sigma(\boldsymbol{w}_{m,1}^{\top} \boldsymbol{x}_i)) \pi_{m,2} \right). \qquad \text{(A.2)}$$

Derivatives are

$$\frac{\partial f^{(1)}(\boldsymbol{x}_i, \boldsymbol{w}, \boldsymbol{\pi})}{\partial \boldsymbol{w}_{m,1}} = \frac{1}{\sqrt{M}} \left(\pi_{m,1} - \pi_{m,2}\right) \boldsymbol{x}_i \dot{\sigma}\left(\boldsymbol{w}_{m,1}^{\top} \boldsymbol{x}_i\right), \qquad \text{(A.3)}$$

$$\frac{\partial f^{(1)}(\boldsymbol{x}_i, \boldsymbol{w}, \boldsymbol{\pi})}{\partial \pi_{m,1}} = \frac{1}{\sqrt{M}} \sigma\left(\boldsymbol{w}_{m,1}^{\top} \boldsymbol{x}_i\right), \qquad \text{(A.4)}$$

$$\frac{\partial f^{(1)}(\boldsymbol{x}_i, \boldsymbol{w}, \boldsymbol{\pi})}{\partial \pi_{m,2}} = -\frac{1}{\sqrt{M}} \left(1 - \sigma\left(\boldsymbol{w}_{m,1}^{\top} \boldsymbol{x}_i\right)\right), \qquad \text{(A.5)}$$

Therefore, since there is only one internal node per a single tree, from the definition of the NTK, the TNTK is obtained as

$$
\begin{aligned}
\widehat{\Theta}^{(1)}\left(\boldsymbol{x}_i, \boldsymbol{x}_j\right) = \sum_{m=1}^{M} & \left( \left\langle \frac{\partial f\left(\boldsymbol{x}_i, \boldsymbol{w}, \boldsymbol{\pi}\right)}{\partial \boldsymbol{w}_{m,1}}, \frac{\partial f\left(\boldsymbol{x}_j, \boldsymbol{w}, \boldsymbol{\pi}\right)}{\partial \boldsymbol{w}_{m,1}} \right\rangle \right. \\
& \left. + \left\langle \frac{\partial f\left(\boldsymbol{x}_i, \boldsymbol{w}, \boldsymbol{\pi}\right)}{\partial \pi_{m,1}}, \frac{\partial f\left(\boldsymbol{x}_j, \boldsymbol{w}, \boldsymbol{\pi}\right)}{\partial \pi_{m,1}} \right\rangle + \left\langle \frac{\partial f\left(\boldsymbol{x}_i, \boldsymbol{w}, \boldsymbol{\pi}\right)}{\partial \pi_{m,2}}, \frac{\partial f\left(\boldsymbol{x}_j, \boldsymbol{w}, \boldsymbol{\pi}\right)}{\partial \pi_{m,2}} \right\rangle \right) \\
= \frac{1}{M} \sum_{m=1}^{M} & \left( (\pi_{m,1} - \pi_{m,2})^2\, \boldsymbol{x}_i^\top \boldsymbol{x}_j \dot{\sigma}(\boldsymbol{w}_{m,1}^\top \boldsymbol{x}_i) \dot{\sigma}(\boldsymbol{w}_{m,1}^\top \boldsymbol{x}_j) + 2\sigma(\boldsymbol{w}_{m,1}^\top \boldsymbol{x}_i)\sigma(\boldsymbol{w}_{m,1}^\top \boldsymbol{x}_j) \right).
\end{aligned}
$$
$$(A.6)$$

Since we are considering the infinite number of trees ($M \to \infty$), the average in Equation (A.6) can be replaced by the expected value by applying the law of the large numbers:

$$
\begin{aligned}
\Theta^{(1)}\left(\boldsymbol{x}_i, \boldsymbol{x}_j\right) = \mathbb{E}_m & \left[ \left( (\pi_{m,1} - \pi_{m,2})^2\, \boldsymbol{x}_i^\top \boldsymbol{x}_j \dot{\sigma}(\boldsymbol{w}_{m,1}^\top \boldsymbol{x}_i) \dot{\sigma}(\boldsymbol{w}_{m,1}^\top \boldsymbol{x}_j) \right. \right. \\
& \left. \left. + 2\sigma(\boldsymbol{w}_{m,1}^\top \boldsymbol{x}_i)\sigma(\boldsymbol{w}_{m,1}^\top \boldsymbol{x}_j) \right) \right] \\
= 2(\Sigma(\boldsymbol{x}_i, \boldsymbol{x}_j)\dot{\mathcal{T}}(\boldsymbol{x}_i, \boldsymbol{x}_j) & + \mathcal{T}(\boldsymbol{x}_i, \boldsymbol{x}_j)),
\end{aligned}
$$
$$(A.7)$$

which is consistent with Equation (7). Here, $\mathbb{E}_m\left[(\pi_{m,1} - \pi_{m,2})^2\right] = 2$ because the variance of $\pi_{m,\ell}$ is 1.0, and $\boldsymbol{w}_{m,1}$ corresponds to $\boldsymbol{u}$ in Theorem 1.

For $d > 1$, we divide the TNTK into four components:

$$
\Theta^{(d)}\left(\boldsymbol{x}_i, \boldsymbol{x}_j\right) = \Theta^{(d),(t)}\left(\boldsymbol{x}_i, \boldsymbol{x}_j\right) + \Theta^{(d),(l)}\left(\boldsymbol{x}_i, \boldsymbol{x}_j\right) + \Theta^{(d),(r)}\left(\boldsymbol{x}_i, \boldsymbol{x}_j\right) + \Theta^{(d),(b)}\left(\boldsymbol{x}_i, \boldsymbol{x}_j\right),
$$

where the indices $(t)$, $(l)$ and $(r)$ mean the parameters of the (t)op of the tree, (l)eft subtree, and (r)ight subtree, respectively. The index $(b)$ implies the (b)ottom of the tree: tree leaves. With Equation (A.7), we have

$$
\Theta^{(1),(t)}\left(\boldsymbol{x}_i, \boldsymbol{x}_j\right) = 2\Sigma(\boldsymbol{x}_i, \boldsymbol{x}_j)\dot{\mathcal{T}}(\boldsymbol{x}_i, \boldsymbol{x}_j), \tag{A.8}
$$
$$
\Theta^{(1),(l)}\left(\boldsymbol{x}_i, \boldsymbol{x}_j\right) = 0, \tag{A.9}
$$
$$
\Theta^{(1),(r)}\left(\boldsymbol{x}_i, \boldsymbol{x}_j\right) = 0, \tag{A.10}
$$
$$
\Theta^{(1),(b)}\left(\boldsymbol{x}_i, \boldsymbol{x}_j\right) = 2\mathcal{T}(\boldsymbol{x}_i, \boldsymbol{x}_j). \tag{A.11}
$$

For each component, we show the following lemmas:

**Lemma 1.**

$$
\Theta^{(d+1),(t)}\left(\boldsymbol{x}_i, \boldsymbol{x}_j\right) = 2\mathcal{T}(\boldsymbol{x}_i, \boldsymbol{x}_j)\Theta^{(d),(t)}\left(\boldsymbol{x}_i, \boldsymbol{x}_j\right) \tag{A.12}
$$

**Lemma 2.**

$$
\begin{aligned}
& \Theta^{(d+1),(l)}\left(\boldsymbol{x}_i, \boldsymbol{x}_j\right) + \Theta^{(d+1),(r)}\left(\boldsymbol{x}_i, \boldsymbol{x}_j\right) \\
& = 2\mathcal{T}(\boldsymbol{x}_i, \boldsymbol{x}_j)(\Theta^{(d),(t)}\left(\boldsymbol{x}_i, \boldsymbol{x}_j\right) + \Theta^{(d),(l)}\left(\boldsymbol{x}_i, \boldsymbol{x}_j\right) + \Theta^{(d),(r)}\left(\boldsymbol{x}_i, \boldsymbol{x}_j\right))
\end{aligned} \tag{A.13}
$$

**Lemma 3.**

$$
\Theta^{(d+1),(b)}\left(\boldsymbol{x}_i, \boldsymbol{x}_j\right) = 2\mathcal{T}(\boldsymbol{x}_i, \boldsymbol{x}_j)\Theta^{(d),(b)}\left(\boldsymbol{x}_i, \boldsymbol{x}_j\right) \tag{A.14}
$$

Combining them, we can derive Equation (7). □

### A.1 PROOF OF LEMMA 1

*Proof.* An incremental formula for the model output with a certain depth tree ensemble is

$$f^{(d)}(\boldsymbol{x}_i, \boldsymbol{w}, \boldsymbol{\pi}) = \frac{1}{\sqrt{M}} \sum_{m=1}^{M} \left( \sigma \left( \boldsymbol{w}_{m,t}^\top \boldsymbol{x}_i \right) f_m^{(d-1)} \left( \boldsymbol{x}_i, \boldsymbol{w}_m^{(l)}, \boldsymbol{\pi}_m^{(l)} \right) \right.$$
$$\left. + \left( 1 - \sigma \left( \boldsymbol{w}_{m,t}^\top \boldsymbol{x}_i \right) \right) f_m^{(d-1)} \left( \boldsymbol{x}_i, \boldsymbol{w}_m^{(r)}, \boldsymbol{\pi}_m^{(r)} \right) \right), \qquad (A.15)$$

where $t$ used in $\boldsymbol{w}_{m,t}$ implies the node at the top of the tree. With Equation (A.15),

$$\frac{\partial f^{(d+1)}(\boldsymbol{x}_i, \boldsymbol{w}, \boldsymbol{\pi})}{\partial \boldsymbol{w}_{m,t}} = \boldsymbol{x}_i \dot{\sigma}(\boldsymbol{w}_{m,t}^\top \boldsymbol{x}_i) \left( f_m^{(d)} \left( \boldsymbol{x}_i, \boldsymbol{w}_m^{(l)}, \boldsymbol{\pi}_m^{(l)} \right) - f_m^{(d)} \left( \boldsymbol{x}_i, \boldsymbol{w}_m^{(r)}, \boldsymbol{\pi}_m^{(r)} \right) \right), \qquad (A.16)$$

$$\widehat{\Theta}^{(d+1),(t)}(\boldsymbol{x}_i, \boldsymbol{x}_j) = \frac{1}{M} \sum_{m=1}^{M} \boldsymbol{x}_i^\top \boldsymbol{x}_j \dot{\sigma}(\boldsymbol{w}_{m,t}^\top \boldsymbol{x}_i) \dot{\sigma}(\boldsymbol{w}_{m,t}^\top \boldsymbol{x}_j) \left( f_m^{(d)} \left( \boldsymbol{x}_i, \boldsymbol{w}_m^{(l)}, \boldsymbol{\pi}_m^{(l)} \right) f_m^{(d)} \left( \boldsymbol{x}_j, \boldsymbol{w}_m^{(l)}, \boldsymbol{\pi}_m^{(l)} \right) \right.$$
$$- f_m^{(d)} \left( \boldsymbol{x}_i, \boldsymbol{w}_m^{(l)}, \boldsymbol{\pi}_m^{(l)} \right) f_m^{(d)} \left( \boldsymbol{x}_j, \boldsymbol{w}_m^{(r)}, \boldsymbol{\pi}_m^{(r)} \right)$$
$$- f_m^{(d)} \left( \boldsymbol{x}_i, \boldsymbol{w}_m^{(r)}, \boldsymbol{\pi}_m^{(r)} \right) f_m^{(d)} \left( \boldsymbol{x}_j, \boldsymbol{w}_m^{(l)}, \boldsymbol{\pi}_m^{(l)} \right)$$
$$\left. + f_m^{(d)} \left( \boldsymbol{x}_i, \boldsymbol{w}_m^{(r)}, \boldsymbol{\pi}_m^{(r)} \right) f_m^{(d)} \left( \boldsymbol{x}_j, \boldsymbol{w}_m^{(r)}, \boldsymbol{\pi}_m^{(r)} \right) \right), \qquad (A.17)$$

Since $f_m^{(d)} \left( \boldsymbol{x}_i, \boldsymbol{w}_m^{(r)}, \boldsymbol{\pi}_m^{(r)} \right)$ and $f_m^{(d)} \left( \boldsymbol{x}_j, \boldsymbol{w}_m^{(l)}, \boldsymbol{\pi}_m^{(l)} \right)$ are independent of each other and have zero-mean Gaussian distribution because of the initialization of $\boldsymbol{\pi}_m$ with zero-mean i.i.d Gaussians[2], $\mathbb{E} \left[ f_m^{(d)} \left( \boldsymbol{x}_i, \boldsymbol{w}_m^{(l)}, \boldsymbol{\pi}_m^{(l)} \right) f_m^{(d)} \left( \boldsymbol{x}_j, \boldsymbol{w}_m^{(r)}, \boldsymbol{\pi}_m^{(r)} \right) \right]$ and $\mathbb{E} \left[ f_m^{(d)} \left( \boldsymbol{x}_i, \boldsymbol{w}_m^{(r)}, \boldsymbol{\pi}_m^{(r)} \right) f_m^{(d)} \left( \boldsymbol{x}_j, \boldsymbol{w}_m^{(l)}, \boldsymbol{\pi}_m^{(l)} \right) \right]$ are zero. Therefore, considering the infinite number of trees $(M \to \infty)$,

$$\Theta^{(d+1),(t)}(\boldsymbol{x}_i, \boldsymbol{x}_j) = \boldsymbol{x}_i^\top \boldsymbol{x}_j \dot{\mathcal{T}}(\boldsymbol{x}_i, \boldsymbol{x}_j) \mathbb{E}_m \left[ f_m^{(d)} \left( \boldsymbol{x}_i, \boldsymbol{w}_m^{(l)}, \boldsymbol{\pi}_m^{(l)} \right) f_m^{(d)} \left( \boldsymbol{x}_j, \boldsymbol{w}_m^{(l)}, \boldsymbol{\pi}_m^{(l)} \right) \right.$$
$$\left. + f_m^{(d)} \left( \boldsymbol{x}_i, \boldsymbol{w}_m^{(r)}, \boldsymbol{\pi}_m^{(r)} \right) f_m^{(d)} \left( \boldsymbol{x}_j, \boldsymbol{w}_m^{(r)}, \boldsymbol{\pi}_m^{(r)} \right) \right] \qquad (A.18)$$

From Equation (A.18), what we need to prove is $\mathbb{E}_m \left[ f_m^{(d)}(\boldsymbol{x}_i, \boldsymbol{w}_m, \boldsymbol{\pi}_m) f_m^{(d)}(\boldsymbol{x}_j, \boldsymbol{w}_m, \boldsymbol{\pi}_m) \right] = (2\mathcal{T}(\boldsymbol{x}_i, \boldsymbol{x}_j))^d$. We do this by induction. In the base case with $d = 1$,

$$\mathbb{E}_m \left[ f_m^{(1)}(\boldsymbol{x}_i, \boldsymbol{w}_m, \boldsymbol{\pi}_m) f_m^{(1)}(\boldsymbol{x}_j, \boldsymbol{w}_m, \boldsymbol{\pi}_m) \right]$$
$$= \mathbb{E}_m \left[ \left( \sigma(\boldsymbol{w}_{m,1}^\top \boldsymbol{x}_i) \pi_{m,1} + \left( 1 - \sigma(\boldsymbol{w}_{m,1}^\top \boldsymbol{x}_i) \right) \pi_{m,2} \right) \left( \sigma(\boldsymbol{w}_{m,1}^\top \boldsymbol{x}_j) \pi_{m,1} + \left( 1 - \sigma(\boldsymbol{w}_{m,1}^\top \boldsymbol{x}_j) \right) \pi_{m,2} \right) \right]$$
$$= \mathbb{E}_m \left[ \underbrace{(\pi_{m,1} - \pi_{m,2})^2}_{\to 2} \sigma(\boldsymbol{w}_{m,1}^\top \boldsymbol{x}_i) \sigma(\boldsymbol{w}_{m,1}^\top \boldsymbol{x}_j) \right.$$
$$\left. + \underbrace{\pi_{m,1} \pi_{m,2}}_{\to 0} (\sigma(\boldsymbol{w}_{m,1}^\top \boldsymbol{x}_i) + \sigma(\boldsymbol{w}_{m,1}^\top \boldsymbol{x}_j)) - \pi_{m,2}^2 (\underbrace{\sigma(\boldsymbol{w}_{m,1}^\top \boldsymbol{x}_i) + \sigma(\boldsymbol{w}_{m,1}^\top \boldsymbol{x}_j) - 1}_{\to 0}) \right]$$
$$= 2\mathcal{T}(\boldsymbol{x}_i, \boldsymbol{x}_j), \qquad (A.19)$$

where we use the property

$$\mathbb{E} \left[ \sigma(v) \right] = 0.5 \qquad (A.20)$$

for any $v$ generated from a zero-mean Gaussian distribution. The subscript arrows $(\to)$ show what the expected value will be.

---

[2]This holds because the model output is a weighted average of $\pi_{m,\ell}$.

Next, when the depth is $d + 1$,

$$\mathbb{E}_m \left[ f_m^{(d+1)}(\boldsymbol{x}_i, \boldsymbol{w}_m, \boldsymbol{\pi}_m) f_m^{(d+1)}(\boldsymbol{x}_j, \boldsymbol{w}_m, \boldsymbol{\pi}_m) \right]$$

$$= \mathbb{E}_m \left[ \left( \sigma(\boldsymbol{w}_{m,t}^\top \boldsymbol{x}_i) f_m^{(d)}\left(\boldsymbol{x}_i, \boldsymbol{w}_m^{(l)}, \boldsymbol{\pi}_m^{(l)}\right) + \left(1 - \sigma(\boldsymbol{w}_{m,t}^\top \boldsymbol{x}_i)\right) f_m^{(d)}\left(\boldsymbol{x}_i, \boldsymbol{w}_m^{(r)}, \boldsymbol{\pi}_m^{(r)}\right) \right) \right.$$

$$\left. \left( \sigma(\boldsymbol{w}_{m,t}^\top \boldsymbol{x}_j) f_m^{(d)}\left(\boldsymbol{x}_j, \boldsymbol{w}_m^{(l)}, \boldsymbol{\pi}_m^{(l)}\right) + \left(1 - \sigma(\boldsymbol{w}_{m,t}^\top \boldsymbol{x}_j)\right) f_m^{(d)}\left(\boldsymbol{x}_j, \boldsymbol{w}_m^{(r)}, \boldsymbol{\pi}_m^{(r)}\right) \right) \right]$$

$$= \mathbb{E}_m \left[ \left( \underbrace{\left( f_m^{(d)}\left(\boldsymbol{x}_i, \boldsymbol{w}_m^{(l)}, \boldsymbol{\pi}_m^{(l)}\right) - f_m^{(d)}\left(\boldsymbol{x}_i, \boldsymbol{w}_m^{(r)}, \boldsymbol{\pi}_m^{(r)}\right) \right) \sigma(\boldsymbol{w}_{m,t}^\top \boldsymbol{x}_i)}_{(A)} + \underbrace{f_m^{(d)}\left(\boldsymbol{x}_i, \boldsymbol{w}_m^{(r)}, \boldsymbol{\pi}_m^{(r)}\right)}_{(B)} \right) \right.$$

$$\left. \left( \underbrace{\left( f_m^{(d)}\left(\boldsymbol{x}_j, \boldsymbol{w}_m^{(l)}, \boldsymbol{\pi}_m^{(l)}\right) - f_m^{(d)}\left(\boldsymbol{x}_j, \boldsymbol{w}_m^{(r)}, \boldsymbol{\pi}_m^{(r)}\right) \right) \sigma(\boldsymbol{w}_{m,t}^\top \boldsymbol{x}_j)}_{(C)} + \underbrace{f_m^{(d)}\left(\boldsymbol{x}_j, \boldsymbol{w}_m^{(r)}, \boldsymbol{\pi}_m^{(r)}\right)}_{(D)} \right) \right],$$

$$\text{(A.21)}$$

where the last equality sign is just a simplification to separate components (A), (B), (C), and (D). Since $f_m^{(d)}(\boldsymbol{x}_i, \boldsymbol{w}_m^{(r)}, \boldsymbol{\pi}_m^{(r)})$ and $f_m^{(d)}(\boldsymbol{x}_j, \boldsymbol{w}_m^{(l)}, \boldsymbol{\pi}_m^{(l)})$ are independent of each other and have zero-mean i.i.d Gaussian distribution, we obtain

$$\mathbb{E}_m\left[(A) \times (C)\right] = \mathcal{T}\left(\boldsymbol{x}_i, \boldsymbol{x}_j\right) \mathbb{E}_m \left[ f_m^{(d)}\left(\boldsymbol{x}_i, \boldsymbol{w}_m^{(l)}, \boldsymbol{\pi}_m^{(l)}\right) f_m^{(d)}\left(\boldsymbol{x}_j, \boldsymbol{w}_m^{(l)}, \boldsymbol{\pi}_m^{(l)}\right) \right.$$

$$\left. + f_m^{(d)}\left(\boldsymbol{x}_i, \boldsymbol{w}_m^{(r)}, \boldsymbol{\pi}_m^{(r)}\right) f_m^{(d)}\left(\boldsymbol{x}_j, \boldsymbol{w}_m^{(r)}, \boldsymbol{\pi}_m^{(r)}\right) \right], \quad \text{(A.22)}$$

$$\mathbb{E}_m\left[(B) \times (C)\right] = -0.5\, \mathbb{E}_m \left[ f_m^{(d)}\left(\boldsymbol{x}_i, \boldsymbol{w}_m^{(r)}, \boldsymbol{\pi}_m^{(r)}\right) f_m^{(d)}\left(\boldsymbol{x}_j, \boldsymbol{w}_m^{(r)}, \boldsymbol{\pi}_m^{(r)}\right) \right], \quad \text{(A.23)}$$

$$\mathbb{E}_m\left[(A) \times (D)\right] = -0.5\, \mathbb{E}_m \left[ f_m^{(d)}\left(\boldsymbol{x}_i, \boldsymbol{w}_m^{(r)}, \boldsymbol{\pi}_m^{(r)}\right) f_m^{(d)}\left(\boldsymbol{x}_j, \boldsymbol{w}_m^{(r)}, \boldsymbol{\pi}_m^{(r)}\right) \right], \quad \text{(A.24)}$$

$$\mathbb{E}_m\left[(B) \times (D)\right] = \mathbb{E}_m \left[ f_m^{(d)}\left(\boldsymbol{x}_i, \boldsymbol{w}_m^{(r)}, \boldsymbol{\pi}_m^{(r)}\right) f_m^{(d)}\left(\boldsymbol{x}_j, \boldsymbol{w}_m^{(r)}, \boldsymbol{\pi}_m^{(r)}\right) \right]. \quad \text{(A.25)}$$

Equation (A.23), Equation (A.24), and Equation (A.25) cancel each other out. Therefore, we obtain

$$\mathbb{E}_m \left[ f_m^{(d+1)}(\boldsymbol{x}_i, \boldsymbol{w}_m, \boldsymbol{\pi}_m) f_m^{(d+1)}(\boldsymbol{x}_j, \boldsymbol{w}_m, \boldsymbol{\pi}_m) \right]$$

$$= 2\mathcal{T}\left(\boldsymbol{x}_i, \boldsymbol{x}_j\right) \mathbb{E}_m \left[ f_m^{(d)}(\boldsymbol{x}_i, \boldsymbol{w}_m, \boldsymbol{\pi}_m) f_m^{(d)}(\boldsymbol{x}_j, \boldsymbol{w}_m, \boldsymbol{\pi}_m) \right]. \quad \text{(A.26)}$$

By induction hypothesis and Equation (A.19), we have $\mathbb{E}_m \left[ f_m^{(d)}(\boldsymbol{x}_i, \boldsymbol{w}_m, \boldsymbol{\pi}_m) f_m^{(d)}(\boldsymbol{x}_j, \boldsymbol{w}_m, \boldsymbol{\pi}_m) \right] = (2\mathcal{T}\left(\boldsymbol{x}_i, \boldsymbol{x}_j\right))^d$. Therefore, the original lemma also follows. $\qquad\square$

## A.2 PROOF OF LEMMA 2

*Proof.* When the depth is $d + 1$, the derivatives of the parameters in the left subtree and the right subtree are

$$\frac{\partial f^{(d+1)}(\boldsymbol{x}_i, \boldsymbol{w}, \boldsymbol{\pi})}{\partial \boldsymbol{w}_{m,l}} = \sigma(\boldsymbol{w}_{m,t}^\top \boldsymbol{x}_i) \frac{\partial f_m^{(d)}\left(\boldsymbol{x}_i, \boldsymbol{w}_m^{(l)}, \boldsymbol{\pi}_m^{(l)}\right)}{\partial \boldsymbol{w}_{m,l}}, \quad \text{(A.27)}$$

$$\frac{\partial f^{(d+1)}(\boldsymbol{x}_i, \boldsymbol{w}, \boldsymbol{\pi})}{\partial \boldsymbol{w}_{m,r}} = \left(1 - \sigma(\boldsymbol{w}_{m,t}^\top \boldsymbol{x}_i)\right) \frac{\partial f_m^{(d)}\left(\boldsymbol{x}_i, \boldsymbol{w}_m^{(r)}, \boldsymbol{\pi}_m^{(r)}\right)}{\partial \boldsymbol{w}_{m,r}}, \quad \text{(A.28)}$$

where $l$ and $r$ used in $\boldsymbol{w}_{m,l}$ and $\boldsymbol{w}_{m,r}$ implies the node at the left subtree and right subtree, respectively. Therefore, the limiting TNTK for the left subtree is

$$\Theta^{(d+1),(l)}\left(\boldsymbol{x}_i, \boldsymbol{x}_j\right) = \mathcal{T}(\boldsymbol{x}_i, \boldsymbol{x}_j)(\Theta^{(d),(t)}\left(\boldsymbol{x}_i, \boldsymbol{x}_j\right) + \Theta^{(d),(l)}\left(\boldsymbol{x}_i, \boldsymbol{x}_j\right) + \Theta^{(d),(r)}\left(\boldsymbol{x}_i, \boldsymbol{x}_j\right)). \quad \text{(A.29)}$$

Similarly, for the right subtree, since

$$\mathbb{E}[(1-\sigma(\boldsymbol{u}^\top\boldsymbol{x}_i)(1-\sigma(\boldsymbol{u}^\top\boldsymbol{x}_j)] = \mathbb{E}\left[1-\underbrace{\sigma(\boldsymbol{u}^\top\boldsymbol{x}_i)}_{\to 0.5}-\underbrace{\sigma(\boldsymbol{u}^\top\boldsymbol{x}_j)}_{\to 0.5}+\sigma(\boldsymbol{u}^\top\boldsymbol{x}_i)\sigma(\boldsymbol{u}^\top\boldsymbol{x}_j)\right]$$
$$= \mathbb{E}[\sigma(\boldsymbol{u}^\top\boldsymbol{x}_i)\sigma(\boldsymbol{u}^\top\boldsymbol{x}_j)], \tag{A.30}$$

we can also derive

$$\Theta^{(d+1),(r)}\left(\boldsymbol{x}_i,\boldsymbol{x}_j\right) = \mathcal{T}(\boldsymbol{x}_i,\boldsymbol{x}_j)(\Theta^{(d),(t)}\left(\boldsymbol{x}_i,\boldsymbol{x}_j\right)+\Theta^{(d),(l)}\left(\boldsymbol{x}_i,\boldsymbol{x}_j\right)+\Theta^{(d),(r)}\left(\boldsymbol{x}_i,\boldsymbol{x}_j\right)). \tag{A.31}$$

Finally, we obtain the following by combining with Equation (A.29) and Equation (A.31),

$$\Theta^{(d+1),(l)}\left(\boldsymbol{x}_i,\boldsymbol{x}_j\right)+\Theta^{(d+1),(r)}\left(\boldsymbol{x}_i,\boldsymbol{x}_j\right)$$
$$= 2\mathcal{T}\left(\boldsymbol{x}_i,\boldsymbol{x}_j\right)\left(\Theta^{(d),(t)}\left(\boldsymbol{x}_i,\boldsymbol{x}_j\right)+\Theta^{(d),(l)}\left(\boldsymbol{x}_i,\boldsymbol{x}_j\right)+\Theta^{(d),(r)}\left(\boldsymbol{x}_i,\boldsymbol{x}_j\right)\right). \tag{A.32}$$

$\square$

## A.3 PROOF OF LEMMA 3

*Proof.* With Equation (1) and Equation (6),

$$\frac{\partial f\left(\boldsymbol{x}_i,\boldsymbol{w},\boldsymbol{\pi}\right)}{\partial \pi_{m,\ell}} = \frac{1}{\sqrt{M}}\mu_{m,\ell}(\boldsymbol{x}_i,\boldsymbol{w}_m)$$
$$= \frac{1}{\sqrt{M}}\prod_{n=1}^{\mathcal{N}}\sigma\left(\boldsymbol{w}_{m,n}^\top\boldsymbol{x}_i\right)^{\mathbb{1}_{\ell\swarrow n}}\left(1-\sigma\left(\boldsymbol{w}_{m,n}^\top\boldsymbol{x}_i\right)\right)^{\mathbb{1}_{n\searrow\ell}}. \tag{A.33}$$

When we focus on a leaf $\ell$, for a tree with depth of $d$, $\mathbb{1}_{n\searrow\ell}$ or $\mathbb{1}_{\ell\swarrow n}$ equals to 1 $d$ times. Therefore, by Equation (A.30), we can say that there is a $\mathcal{T}(\boldsymbol{x}_i,\boldsymbol{x}_j)^d$ contribution to the limiting kernel $\Theta^{(d),(b)}\left(\boldsymbol{x}_i,\boldsymbol{x}_j\right)$ per leaf index $\ell\in[\mathcal{L}]$. Since there are $2^d$ leaf indices in the perfect binary tree,

$$\Theta^{(d+1),(b)}\left(\boldsymbol{x}_i,\boldsymbol{x}_j\right) = (2\mathcal{T}(\boldsymbol{x}_i,\boldsymbol{x}_j))^d. \tag{A.34}$$

In other words, since the number of leaves doubles with each additional depth, we can say

$$\Theta^{(d+1),(b)}\left(\boldsymbol{x}_i,\boldsymbol{x}_j\right) = 2\mathcal{T}(\boldsymbol{x}_i,\boldsymbol{x}_j)\Theta^{(d),(b)}\left(\boldsymbol{x}_i,\boldsymbol{x}_j\right). \tag{A.35}$$

$\square$

## A.4 CLOSED-FORM FORMULA FOR THE SCALED ERROR FUNCTION

Since we are using the scaled error function as a decision function defined as

$$g_{m,n}(\boldsymbol{w}_{m,n},\boldsymbol{x}_i) = \sigma\left(\boldsymbol{w}_{m,n}^\top\boldsymbol{x}_i\right)$$
$$= \frac{1}{2}\operatorname{erf}\left(\alpha\boldsymbol{w}_{m,n}^\top\boldsymbol{x}_i\right)+\frac{1}{2}, \tag{A.36}$$

$\mathcal{T}$ and $\dot{\mathcal{T}}$ in Theorem 1 can be calculated analytically. Closed-form solutions for the error function (Williams, 1996; Lee et al., 2019) are known to be

$$\mathcal{T}_{\mathrm{erf}}(\boldsymbol{x}_i,\boldsymbol{x}_j) := \mathbb{E}[\operatorname{erf}(\boldsymbol{u}^\top\boldsymbol{x}_i)\operatorname{erf}(\boldsymbol{u}^\top\boldsymbol{x}_j)] = \frac{2}{\pi}\arcsin\left(\frac{\Sigma(\boldsymbol{x}_i,\boldsymbol{x}_j)}{\sqrt{(\Sigma(\boldsymbol{x}_i,\boldsymbol{x}_i)+0.5)(\Sigma(\boldsymbol{x}_j,\boldsymbol{x}_j)+0.5)}}\right),$$
$$\tag{A.37}$$

$$\dot{\mathcal{T}}_{\mathrm{erf}}(\boldsymbol{x}_i,\boldsymbol{x}_j) := \mathbb{E}[\dot{\operatorname{erf}}(\boldsymbol{u}^\top\boldsymbol{x}_i)\dot{\operatorname{erf}}(\boldsymbol{u}^\top\boldsymbol{x}_j)] = \frac{4}{\pi}\frac{1}{\sqrt{(1+2\Sigma(\boldsymbol{x}_i,\boldsymbol{x}_i))(1+2\Sigma(\boldsymbol{x}_j,\boldsymbol{x}_j))-4\Sigma(\boldsymbol{x}_i,\boldsymbol{x}_j)^2}}.$$
$$\tag{A.38}$$

Using the above equations, we can calculate $\mathcal{T}$ and $\dot{\mathcal{T}}$ with the scaled error function as

$$
\begin{aligned}
\mathcal{T}(\boldsymbol{x}_i, \boldsymbol{x}_j) &= \mathbb{E}\left[\frac{1}{4}\operatorname{erf}(\alpha\boldsymbol{u}^\top\boldsymbol{x}_i)\operatorname{erf}(\alpha\boldsymbol{u}^\top\boldsymbol{x}_j)\right] + \mathbb{E}\left[\frac{1}{4}\operatorname{erf}(\alpha\boldsymbol{u}^\top\boldsymbol{x}_i) + \frac{1}{4}\operatorname{erf}(\alpha\boldsymbol{u}^\top\boldsymbol{x}_j)\right] + \frac{1}{4} \\
&= \frac{1}{4}\mathbb{E}\left[\operatorname{erf}(\alpha\boldsymbol{u}^\top\boldsymbol{x}_i)\operatorname{erf}(\alpha\boldsymbol{u}^\top\boldsymbol{x}_j)\right] + \frac{1}{4} \\
&= \frac{1}{2\pi}\arcsin\left(\frac{\alpha^2\Sigma(\boldsymbol{x}_i, \boldsymbol{x}_j)}{\sqrt{(\alpha^2\Sigma(\boldsymbol{x}_i, \boldsymbol{x}_i) + 0.5)(\alpha^2\Sigma(\boldsymbol{x}_j, \boldsymbol{x}_j) + 0.5)}}\right) + \frac{1}{4}, \quad\quad\quad (A.39) \\
\dot{\mathcal{T}}(\boldsymbol{x}_i, \boldsymbol{x}_j) &= \frac{\alpha^2}{4}\mathbb{E}\left[\dot{\operatorname{erf}}(\alpha\boldsymbol{u}^\top\boldsymbol{x}_i)\dot{\operatorname{erf}}(\alpha\boldsymbol{u}^\top\boldsymbol{x}_j)\right] \\
&= \frac{\alpha^2}{\pi}\frac{1}{\sqrt{(1 + 2\alpha^2\Sigma(\boldsymbol{x}_i, \boldsymbol{x}_i))(1 + 2\alpha^2\Sigma(\boldsymbol{x}_j, \boldsymbol{x}_j)) - 4\alpha^4\Sigma(\boldsymbol{x}_i, \boldsymbol{x}_j)^2}}. \quad\quad (A.40)
\end{aligned}
$$

# B  NEURAL TANGENT KERNEL FOR MULTI-LAYER PERCEPTRON

## B.1  EQUIVALENCE BETWEEN THE TWO-LAYER PERCEPTRON AND TREES OF DEPTH 1

In the following, we describe a two-layer perceptron using the same symbols used in soft trees (Section 2.1) to make it easier to see the correspondences between a two-layer perceptron and a soft tree ensemble. A two-layer perceptron is given as

$$
f_{\mathrm{MLP}\,(2)}(\boldsymbol{x}_i, \boldsymbol{w}, \boldsymbol{a}) = \frac{1}{\sqrt{M}}\sum_{m=1}^{M} a_m\sigma\left(\boldsymbol{w}_m^\top\boldsymbol{x}_i\right), \quad\quad\quad (B.1)
$$

where we use $M$ as the number of the hidden layer nodes, $\sigma$ as a nonlinear activation function, and $\boldsymbol{w} = (\boldsymbol{w}_1, \ldots, \boldsymbol{w}_M) \in \mathbb{R}^{F \times M}$ and $\boldsymbol{a} = (a_1, \ldots, a_M) \in \mathbb{R}^{1 \times M}$ as parameters at the first and second layers initialized by zero-mean Gaussians with unit variances. Since

$$
\frac{\partial f_{\mathrm{MLP}\,(2)}(\boldsymbol{x}_i, \boldsymbol{w}, \boldsymbol{a})}{\partial \boldsymbol{w}_m} = \frac{1}{\sqrt{M}} a_m\boldsymbol{x}_i\dot{\sigma}\left(\boldsymbol{w}_m^\top\boldsymbol{x}_i\right), \quad\quad\quad (B.2)
$$

$$
\frac{\partial f_{\mathrm{MLP}\,(2)}(\boldsymbol{x}_i, \boldsymbol{w}, \boldsymbol{a})}{\partial a_m} = \frac{1}{\sqrt{M}}\sigma\left(\boldsymbol{w}_m^\top\boldsymbol{x}_i\right), \quad\quad\quad (B.3)
$$

we have

$$
\widehat{\Theta}^{\mathrm{MLP}\,(2)}(\boldsymbol{x}_i, \boldsymbol{x}_j) = \frac{1}{M}\sum_{m=1}^{M}\left(\underbrace{a_m^2\boldsymbol{x}_i^\top\boldsymbol{x}_j\dot{\sigma}\left(\boldsymbol{w}_m^\top\boldsymbol{x}_i\right)\dot{\sigma}\left(\boldsymbol{w}_m^\top\boldsymbol{x}_j\right)}_{\text{contribution from the first layer}} + \underbrace{\sigma\left(\boldsymbol{w}_m^\top\boldsymbol{x}_i\right)\sigma\left(\boldsymbol{w}_m^\top\boldsymbol{x}_j\right)}_{\text{contribution from the second layer}}\right).
$$
$$(B.4)$$

Considering the infinite width limit ($M \to \infty$), we have

$$
\Theta^{\mathrm{MLP}\,(2)}(\boldsymbol{x}_i, \boldsymbol{x}_j) = \Sigma(\boldsymbol{x}_i, \boldsymbol{x}_j)\dot{\mathcal{T}}(\boldsymbol{x}_i, \boldsymbol{x}_j) + \mathcal{T}(\boldsymbol{x}_i, \boldsymbol{x}_j), \quad\quad\quad (B.5)
$$

which is the same as the limiting TNTK shown in Equation (7) with $d = 1$ up to constant multiple.

## B.2  FORMULA FOR THE MLP-INDUCED NTK

Based on Arora et al. (2019), we defined the $L$-hidden-layer perceptron[3] as

$$
f_{\mathrm{MLP}\,(\mathrm{L})}(\boldsymbol{x}_i, \boldsymbol{W}) = \boldsymbol{W}^{(L+1)} \cdot \frac{1}{\sqrt{M_L}}\sigma\left(\boldsymbol{W}^{(L)} \cdot \frac{1}{\sqrt{M_{L-1}}}\sigma\left(\boldsymbol{W}^{(L-1)}\cdots\frac{1}{\sqrt{M_1}}\sigma\left(\boldsymbol{W}^{(1)}\boldsymbol{x}_i\right)\right)\right),
$$
$$(B.6)$$

where $\boldsymbol{W}^{(1)} \in \mathbb{R}^{M_1 \times F}$, $\boldsymbol{W}^{(h)} \in \mathbb{R}^{M_h \times M_{h-1}}$, and $\boldsymbol{W}^{(L+1)} \in \mathbb{R}^{1 \times M_L}$ are trainable parameters. We initialize all the weights $\boldsymbol{W} = \left(\boldsymbol{W}^{(1)}, \ldots, \boldsymbol{W}^{(L+1)}\right)$ to values independently drawn from the

---

[3]Note that the two-layer perceptron is the single-hidden-layer perceptron.

standard normal distribution. Considering the limit of the infinite width $M_1, M_2, \ldots, M_L \to \infty$, the formula for the limiting NTK of $L$-hidden-layer MLP is known to be

$$\Theta^{\mathrm{MLP}(L)}\left(\boldsymbol{x}_i, \boldsymbol{x}_j\right) = \sum_{h=1}^{L+1} \left( \Sigma^{(h-1)}\left(\boldsymbol{x}_i, \boldsymbol{x}_j\right) \cdot \prod_{h'=h}^{L+1} \dot{\Sigma}^{(h')}\left(\boldsymbol{x}_i, \boldsymbol{x}_j\right) \right), \tag{B.7}$$

where

$$\Sigma^{(0)}\left(\boldsymbol{x}_i, \boldsymbol{x}_j\right) := \boldsymbol{x}_i^{\top} \boldsymbol{x}_j, \tag{B.8}$$

$$\boldsymbol{\Lambda}^{(h)}\left(\boldsymbol{x}_i, \boldsymbol{x}_j\right) := \begin{pmatrix} \Sigma^{(h-1)}(\boldsymbol{x}_i, \boldsymbol{x}_i) & \Sigma^{(h-1)}(\boldsymbol{x}_i, \boldsymbol{x}_j) \\ \Sigma^{(h-1)}(\boldsymbol{x}_j, \boldsymbol{x}_i) & \Sigma^{(h-1)}(\boldsymbol{x}_j, \boldsymbol{x}_j) \end{pmatrix} \in \mathbb{R}^{2 \times 2}, \tag{B.9}$$

$$\Sigma^{(h)}\left(\boldsymbol{x}_i, \boldsymbol{x}_j\right) := \mathbb{E}_{u,v \sim \mathrm{Normal}\left(0, \boldsymbol{\Lambda}^{(h)}(\boldsymbol{x}_i, \boldsymbol{x}_j)\right)} \left[ \sigma(u)\sigma(v) \right], \tag{B.10}$$

$$\dot{\Sigma}^{(h)}\left(\boldsymbol{x}_i, \boldsymbol{x}_j\right) := \mathbb{E}_{u,v \sim \mathrm{Normal}\left(0, \boldsymbol{\Lambda}^{(h)}(\boldsymbol{x}_i, \boldsymbol{x}_j)\right)} \left[ \dot{\sigma}(u)\dot{\sigma}(v) \right]. \tag{B.11}$$

We let $\dot{\Sigma}^{(L+1)}\left(\boldsymbol{x}_i, \boldsymbol{x}_j\right) := 1$ for convenience. See Arora et al. (2019) for derivation. There is a correspondence between $\Sigma^{(1)}\left(\boldsymbol{x}_i, \boldsymbol{x}_j\right)$ and $\Sigma\left(\boldsymbol{x}_i, \boldsymbol{x}_j\right)$ in Theorem 1, $\Sigma^{(1)}\left(\boldsymbol{x}_i, \boldsymbol{x}_j\right)$ and $\mathcal{T}\left(\boldsymbol{x}_i, \boldsymbol{x}_j\right)$ in Theorem 1, and $\dot{\Sigma}^{(1)}\left(\boldsymbol{x}_i, \boldsymbol{x}_j\right)$ and $\dot{\mathcal{T}}\left(\boldsymbol{x}_i, \boldsymbol{x}_j\right)$ in Theorem 1, respectively.

Since the recursive calculation is needed in Equation (B.7), the computational cost increases as the layers get deeper. It can be seen that the effect of increasing depth is different from that of the limiting TNTK, in which the depth of the tree affects only the value of the exponential power as shown in Equation (7). Therefore, for any tree depth larger than 1, the limiting NTK induced by the MLP with any number of layers and the limiting TNTK do not match.

## C   PROOF OF PROPOSITION 1

*Proof.* As shown in Section 4.1.1, there is a close correspondence between the soft tree ensemble of depth 1 and the two-layer perceptron. On one hand, from Equation (A.7), the limiting TNTK induced by infinite trees with the depth of 1 is $2(\Sigma(\boldsymbol{x}_i, \boldsymbol{x}_j)\dot{\mathcal{T}}(\boldsymbol{x}_i, \boldsymbol{x}_j) + \mathcal{T}(\boldsymbol{x}_i, \boldsymbol{x}_j))$. On the other hand, if the activation function used in the two-layer perceptron is same as $\sigma$ defined in Equation (5), the NTK induced by the infinite width two-layer MLP is $\Sigma(\boldsymbol{x}_i, \boldsymbol{x}_j)\dot{\mathcal{T}}(\boldsymbol{x}_i, \boldsymbol{x}_j) + \mathcal{T}(\boldsymbol{x}_i, \boldsymbol{x}_j)$ (Jacot et al., 2018; Lee et al., 2019). Hence these are exactly the same kernel up to constant multiple.

The conditions under which the MLP-induced NTK are positively definite have already been studied.

**Lemma 4** (Jacot et al. (2018)). *For a non-polynomial Lipschitz nonlinearity $\sigma$, for any input dimension $F$, the NTK induced by the infinite width MLP is positive definite if $\|\boldsymbol{x}_i\|_2 = 1$ for all $i \in [N]$ and $\boldsymbol{x}_i \neq \boldsymbol{x}_j$ $(i \neq j)$.*

Note that $\sigma$ defined in Equation (5) has the non-polynomial Lipschitz nonlinearity. Since the positive definite kernel multiplied by a constant is a positive definite kernel, it follows that the limiting TNTK $\Theta^{(1)}(\boldsymbol{x}_i, \boldsymbol{x}_j)$ for the depth 1 is also positive definite.

As shown in Equation (C.1), as the trees get deeper, $\mathcal{T}(\boldsymbol{x}_i, \boldsymbol{x}_j)$ defined in Equation (8) is multiplied multiple times in the limiting TNTK:

$$\Theta^{(d)}(\boldsymbol{x}_i, \boldsymbol{x}_j) = \underbrace{2^d d \, \Sigma(\boldsymbol{x}_i, \boldsymbol{x}_j)(\mathcal{T}(\boldsymbol{x}_i, \boldsymbol{x}_j))^{d-1}\dot{\mathcal{T}}(\boldsymbol{x}_i, \boldsymbol{x}_j)}_{\text{contribution from inner nodes}} + \underbrace{(2\mathcal{T}(\boldsymbol{x}_i, \boldsymbol{x}_j))^d}_{\text{contribution from leaves}}$$

$$= 2(2\mathcal{T}(\boldsymbol{x}_i, \boldsymbol{x}_j))^{d-1} \underbrace{(d \, \Sigma(\boldsymbol{x}_i, \boldsymbol{x}_j)\dot{\mathcal{T}}(\boldsymbol{x}_i, \boldsymbol{x}_j) + \mathcal{T}(\boldsymbol{x}_i, \boldsymbol{x}_j))}_{\text{NTK induced by two-layer perceptron (if } d = 1)}. \tag{C.1}$$

The positive definiteness of $\mathcal{T}(\boldsymbol{x}_i, \boldsymbol{x}_j)$ has already been proven.

**Lemma 5** (Jacot et al. (2018)). *For a non-polynomial Lipschitz nonlinearity $\sigma$, for any input dimension $F$, the $\mathcal{T}(\boldsymbol{x}_i, \boldsymbol{x}_j) := \mathbb{E}[\sigma(\boldsymbol{u}^{\top} \boldsymbol{x}_i)\sigma(\boldsymbol{u}^{\top} \boldsymbol{x}_j)]$ defined in Theorem 1 is positive definite if $\|\boldsymbol{x}_i\|_2 = 1$ for all $i \in [N]$ and $\boldsymbol{x}_i \neq \boldsymbol{x}_j$ $(i \neq j)$.*

Note that $d \, \Sigma(\boldsymbol{x}_i, \boldsymbol{x}_j)\dot{\mathcal{T}}(\boldsymbol{x}_i, \boldsymbol{x}_j) + \mathcal{T}(\boldsymbol{x}_i, \boldsymbol{x}_j)$ for $d \in \mathbb{N}$ is positive definite. Since the product of the positive definite kernel is positive definite, for infinite trees of arbitrary depth, the positive definiteness of $\Theta^{(d)}(\boldsymbol{x}_i, \boldsymbol{x}_j)$ holds under the same conditions as in MLP. $\square$

# D   PROOF OF THEOREM 2

*Proof.* We use the following lemmas in the proof.

**Lemma 6.** *Let $\boldsymbol{a} \in \mathbb{R}^n$ be a random vector whose entries are independent standard normal random variables. For every $v \geq 0$, with probability at least $1 - 2^n e^{(-v^2 n/2)}$ we have:*

$$\|\boldsymbol{a}\|_1 \leq vn. \tag{D.1}$$

**Lemma 7.** *Let $a_i \in \mathbb{R}_{\geq 0}$. We have*

$$\sum_{i=1}^{n} \sqrt{a_i} \leq \sqrt{n}\sqrt{\sum_{i=1}^{n} a_i}. \tag{D.2}$$

In addition, our proof is based on the strategy used in Lee et al. (2019), which relies on the local Lipschitzness of the model Jacobian at initialization $\boldsymbol{J}(\boldsymbol{x}, \boldsymbol{\theta})$, whose $(i, j)$ entry is $\frac{\partial f(\boldsymbol{x}_i, \boldsymbol{\theta})}{\partial \theta_j}$ where $\theta_j$ is a $j$-th component of $\boldsymbol{\theta}$:

**Theorem 4** (Lee et al. (2019)). *Assume that the limiting NTK induced by any model architecture is positive definite for input sets $\boldsymbol{x}$, such that minimum eigenvalue of the NTK $\lambda_{\min} > 0$. For models with local Lipschitz Jacobian trained under gradient flow with a learning rate $\eta < 2(\lambda_{\min} + \lambda_{\max})$, we have with high probability:*

$$\sup \left| \widehat{\Theta}_{\tau}^*(\boldsymbol{x}_i, \boldsymbol{x}_j) - \widehat{\Theta}_0^*(\boldsymbol{x}_i, \boldsymbol{x}_j) \right| = \mathcal{O}\left(\frac{1}{\sqrt{M}}\right). \tag{D.3}$$

It is not obvious whether or not the soft tree ensemble's Jacobian is local Lipschitz. Therefore, we prove Lemma 8 to prove Theorem 2.

**Lemma 8.** *For soft tree ensemble models with the NTK initialization and a positive finite scaling factor $\alpha$, there is $K > 0$ such that for every $C > 0$, with high probability, the following holds:*

$$\begin{cases} \|\boldsymbol{J}(\boldsymbol{x}, \boldsymbol{\theta})\|_F & \leq K \\ \|\boldsymbol{J}(\boldsymbol{x}, \boldsymbol{\theta}) - \boldsymbol{J}(\boldsymbol{x}, \tilde{\boldsymbol{\theta}})\|_F & \leq K\|\boldsymbol{\theta} - \tilde{\boldsymbol{\theta}}\|_2 \end{cases} , \forall \boldsymbol{\theta}, \tilde{\boldsymbol{\theta}} \in B(\boldsymbol{\theta}_0, C) , \tag{D.4}$$

*where*

$$B(\theta_0, C) := \{\boldsymbol{\theta} : \|\boldsymbol{\theta} - \boldsymbol{\theta}_0\|_2 < C\}. \tag{D.5}$$

By proving that the soft tree ensemble's Jacobian under the NTK initialization is the local Lipschitz with high probability, we extend Theorem 2 for the TNTK. $\qquad \square$

## D.1   PROOF OF LEMMA 6

*Proof.* By use of the Chebyshev's inequality, for some constant $c$, we obtain

$$P(\|\boldsymbol{a}\|_1 > c) \leq \mathbb{E}[e^{\gamma\|\boldsymbol{a}\|_1}]/e^{\gamma c}$$
$$= \left(e^{\gamma^2/2}(1 + \mathrm{erf}(\gamma/\sqrt{2}))\right)^n / e^{\gamma c}, \tag{D.6}$$

where $P$ means a probability. Since $\mathrm{erf}(\gamma/\sqrt{2}) \leq 1$, when we use $\gamma = c/n$, we get

$$P(\|\boldsymbol{a}\|_1 > c) \leq 2^n e^{(-c^2/2n)}. \tag{D.7}$$

Lemma 6 can be obtained by assigning $vn$ to $c$. $\qquad \square$

Figure 8 shows the right-hand side of the Equation (D.7) with $c = 5n$. when $n = 1$, probability is $7.45 \times 10^{-6}$. As $n$ becomes larger, the probability becomes even smaller.

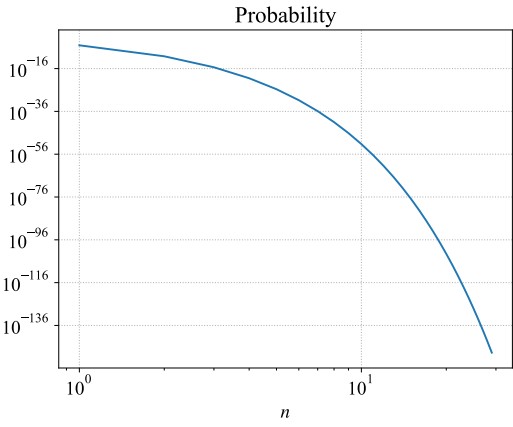

Figure 8: Right-hand side of the Equation (D.7), where $c = 5n$ (in other words, $v = 5$ in Lemma 6).

## D.2 PROOF OF LEMMA 7

*Proof.* By use of Cauchy-Schwarz inequality, for $p, q, x, y \in \mathbb{R}_{\geq 0}$, we have

$$p\sqrt{x} + q\sqrt{y} \leq \sqrt{(p^2 + q^2)(x + y)}. \tag{D.8}$$

With Equation (D.8), we prove the lemma by induction. In the base case,

$$\sqrt{a_1} + \sqrt{a_2} \leq \sqrt{2}\sqrt{a_1 + a_2}, \tag{D.9}$$

which is consistent to the lemma. Next, when we assume

$$\sqrt{a_1} + \cdots + \sqrt{a_k} \leq \sqrt{k}\sqrt{a_1 + \cdots a_k}, \tag{D.10}$$

we have

$$\begin{aligned}
\sqrt{a_1} + \cdots + \sqrt{a_k} + \sqrt{a_{k+1}} &= \left(\sqrt{a_1} + \cdots + \sqrt{a_k}\right) + \sqrt{a_{k+1}} \\
&\leq \sqrt{k}\sqrt{a_1 + \cdots + a_k} + \sqrt{a_{k+1}} \\
&\leq \sqrt{k+1}\sqrt{a_1 + \cdots + a_k + a_{k+1}}.
\end{aligned} \tag{D.11}$$

$\square$

## D.3 PROOF OF LEMMA 8

*Proof.* Consider the contribution of the leaf parameters at first:

$$\frac{\partial f(\boldsymbol{x}_i, \boldsymbol{w}, \boldsymbol{\pi})}{\partial \pi_{m,\ell}} = \frac{1}{\sqrt{M}}\mu_{m,\ell}(\boldsymbol{x}_i, \boldsymbol{w}_m). \tag{D.12}$$

Next, the contribution from the leaf parameters is

$$\begin{aligned}
\frac{\partial f(\boldsymbol{x}_i, \boldsymbol{w}, \boldsymbol{\pi})}{\partial \boldsymbol{w}_{m,n}} &= \frac{1}{\sqrt{M}}\sum_{\ell=1}^{\mathcal{L}} \pi_{m,\ell}\frac{\partial \mu_{m,\ell}(\boldsymbol{x}_i, \boldsymbol{w}_m)}{\partial \boldsymbol{w}_{m,n}} \\
&= \frac{1}{\sqrt{M}}\sum_{\ell=1}^{\mathcal{L}} \pi_{m,\ell}S_{n,\ell}(\boldsymbol{x}_i, \boldsymbol{w}_m)\boldsymbol{x}_i\dot{\sigma}\left(\boldsymbol{w}_{m,n}^{\top}\boldsymbol{x}_i\right),
\end{aligned} \tag{D.13}$$

where

$$S_{n,\ell}(\boldsymbol{x}, \boldsymbol{w}_m) := \left(\prod_{n'=1}^{\mathcal{N}} \sigma\left(\boldsymbol{w}_{m,n'}^{\top}\boldsymbol{x}_i\right)^{\mathbb{1}_{(\ell\swarrow n')\&(n\neq n')}}\left(1 - \sigma\left(\boldsymbol{w}_{m,n'}^{\top}\boldsymbol{x}_i\right)\right)^{\mathbb{1}_{(n'\searrow \ell)\&(n\neq n')}}\right)(-1)^{\mathbb{1}_{n\searrow \ell}}, \tag{D.14}$$

and $\&$ is a logical conjunction. For any real scalar $p$ and $q$, the scaled error function $\sigma$ defined in Equation (5) is bounded as follows:

$$0 \le \sigma(p) \le 1, \quad |\sigma(p) - \sigma(q)| \le |p - q|, \quad 0 \le \dot{\sigma}(p) \le \alpha, \quad |\dot{\sigma}(p) - \dot{\sigma}(q)| \le \alpha|p - q|. \quad \text{(D.15)}$$

Therefore, the absolute value of $S_{n,\ell}$ does not exceed 1. With Equation (D.15), we can obtain

$$\left\| \frac{\partial \mu_{m,\ell}(\boldsymbol{x}_i, \boldsymbol{w}_m)}{\partial \boldsymbol{w}_{m,n}} \right\|_2 = \left\| S_{n,\ell}(\boldsymbol{x}_i, \boldsymbol{w}_m)\boldsymbol{x}_i \dot{\sigma}\left(\boldsymbol{w}_{m,n}^\top \boldsymbol{x}_i\right) \right\|_2 \le \alpha \|\boldsymbol{x}_i\|_2 \quad \text{(D.16)}$$

with high probability. Therefore, with Lemma 6, in probability,

$$
\begin{aligned}
\|\boldsymbol{J}(\boldsymbol{x}, \boldsymbol{\theta})\|_F^2 &= \sum_{i=1}^{N} \left( \|\boldsymbol{J}(\boldsymbol{x}_i, \boldsymbol{w})\|_F^2 + \|\boldsymbol{J}(\boldsymbol{x}_i, \boldsymbol{\pi})\|_F^2 \right) \\
&= \frac{1}{M} \sum_{i=1}^{N} \sum_{m=1}^{M} \left( \sum_{n=1}^{\mathcal{N}} \left( \left\| \sum_{\ell=1}^{\mathcal{L}} \pi_{m,\ell} \frac{\partial \mu_{m,\ell}(\boldsymbol{x}_i, \boldsymbol{w}_m)}{\partial \boldsymbol{w}_{m,n}} \right\|_2^2 \right) + \sum_{\ell=1}^{\mathcal{L}} \left( \mu_{m,\ell}(\boldsymbol{x}_i, \boldsymbol{w}_m)^2 \right) \right) \\
&\le \frac{1}{M} \sum_{i=1}^{N} \sum_{m=1}^{M} \left( \sum_{n=1}^{\mathcal{N}} v^2 \mathcal{L}^2 \alpha^2 \|\boldsymbol{x}_i\|_2^2 + \sum_{\ell=1}^{\mathcal{L}} 1 \right) \\
&= \sum_{i=1}^{N} \mathcal{L}(v^2 \mathcal{L} \alpha^2 \mathcal{N} \|\boldsymbol{x}_i\|_2^2 + 1). \quad \text{(D.17)}
\end{aligned}
$$

Next, we will consider the Jacobian difference. Since $\mu_{m,\ell}(\boldsymbol{x}_i, \boldsymbol{w}_m)$ is a multiple multiplication of the decision function, by use of

$$\left| \prod_{i=1}^{n} p_i - \prod_{i=1}^{n} q_i \right| \le \sum_{i=1}^{n} |p_i - q_i| \quad \text{for } |p_i|, |q_i| \le 1, \quad \text{(D.18)}$$

we obtain

$$
\begin{aligned}
|\mu_{m,\ell}(\boldsymbol{x}_i, \boldsymbol{w}_m) - \mu_{m,\ell}(\boldsymbol{x}_i, \tilde{\boldsymbol{w}}_m)| &= \left| \prod_{n=1}^{\mathcal{N}} \sigma\left(\boldsymbol{w}_{m,n}^\top \boldsymbol{x}_i\right)^{\mathbb{1}_{\ell \swarrow n}} \left(1 - \sigma\left(\boldsymbol{w}_{m,n}^\top \boldsymbol{x}_i\right)\right)^{\mathbb{1}_{n \searrow \ell}} \right. \\
&\qquad \left. - \prod_{n=1}^{\mathcal{N}} \sigma\left(\tilde{\boldsymbol{w}}_{m,n}^\top \boldsymbol{x}_i\right)^{\mathbb{1}_{\ell \swarrow n}} \left(1 - \sigma\left(\tilde{\boldsymbol{w}}_{m,n}^\top \boldsymbol{x}_i\right)\right)^{\mathbb{1}_{n \searrow \ell}} \right| \\
&\le \sum_{n=1}^{\mathcal{N}} \left| \sigma\left(\boldsymbol{w}_{m,n}^\top \boldsymbol{x}_i\right)^{\mathbb{1}_{\ell \swarrow n}} \left(1 - \sigma\left(\boldsymbol{w}_{m,n}^\top \boldsymbol{x}_i\right)\right)^{\mathbb{1}_{n \searrow \ell}} \right. \\
&\qquad \left. - \sigma\left(\tilde{\boldsymbol{w}}_{m,n}^\top \boldsymbol{x}_i\right)^{\mathbb{1}_{\ell \swarrow n}} \left(1 - \sigma\left(\tilde{\boldsymbol{w}}_{m,n}^\top \boldsymbol{x}_i\right)\right)^{\mathbb{1}_{n \searrow \ell}} \right| \\
&\le \sum_{n=1}^{\mathcal{N}} |\boldsymbol{w}_{m,n}^\top \boldsymbol{x}_i - \tilde{\boldsymbol{w}}_{m,n}^\top \boldsymbol{x}_i| \\
&\le \sum_{n=1}^{\mathcal{N}} \|\boldsymbol{x}_i\|_2 \|\boldsymbol{w}_{m,n} - \tilde{\boldsymbol{w}}_{m,n}\|_2, \quad \text{(D.19)}
\end{aligned}
$$

where it should be noted that $(\ell \swarrow n) \& (n \searrow \ell)$ must be false.

$S_{n,\ell}\left(\boldsymbol{x}_i, \boldsymbol{w}_m\right) - S_{n,\ell}\left(\boldsymbol{x}_i, \tilde{\boldsymbol{w}}_m\right)$ can be bound in the same way as Equation (D.19). Therefore, we also obtain

$$
\begin{aligned}
\left\| \frac{\partial \mu_{m,\ell}(\boldsymbol{x}_i, \boldsymbol{w}_m)}{\partial \boldsymbol{w}_{m,n}} - \frac{\partial \mu_{m,\ell}(\boldsymbol{x}_i, \tilde{\boldsymbol{w}}_m)}{\partial \tilde{\boldsymbol{w}}_{m,n}} \right\|_2 &= \| S_{n,\ell}\left(\boldsymbol{x}_i, \boldsymbol{w}_m\right) \boldsymbol{x}_i \dot{\sigma}\left(\boldsymbol{w}_{m,n}^\top \boldsymbol{x}_i\right) - S_{n,\ell}\left(\boldsymbol{x}_i, \tilde{\boldsymbol{w}}_m\right) \boldsymbol{x}_i \dot{\sigma}\left(\tilde{\boldsymbol{w}}_{m,n}^\top \boldsymbol{x}_i\right) \|_2 \\
&= \|\boldsymbol{x}_i\|_2 | S_{n,\ell}\left(\boldsymbol{x}_i, \boldsymbol{w}_m\right) \dot{\sigma}\left(\boldsymbol{w}_{m,n}^\top \boldsymbol{x}_i\right) - S_{n,\ell}\left(\boldsymbol{x}_i, \tilde{\boldsymbol{w}}_m\right) \dot{\sigma}\left(\tilde{\boldsymbol{w}}_{m,n}^\top \boldsymbol{x}_i\right) | \\
&\leq \|\boldsymbol{x}_i\|_2 \Big( |(S_{n,\ell}\left(\boldsymbol{x}_i, \boldsymbol{w}_m\right) - S_{n,\ell}\left(\boldsymbol{x}_i, \tilde{\boldsymbol{w}}_m\right)) \dot{\sigma}\left(\boldsymbol{w}_{m,n}^\top \boldsymbol{x}_i\right) | \\
&\qquad\qquad + |(\dot{\sigma}\left(\boldsymbol{w}_{m,n}^\top \boldsymbol{x}_i\right) - \dot{\sigma}\left(\tilde{\boldsymbol{w}}_{m,n}^\top \boldsymbol{x}_i\right)) S_{n,\ell}\left(\boldsymbol{x}_i, \tilde{\boldsymbol{w}}_m\right) | \Big) \\
&\leq \|\boldsymbol{x}_i\|_2 \Big( |\alpha(S_{n,\ell}\left(\boldsymbol{x}_i, \boldsymbol{w}_m\right) - S_{n,\ell}\left(\boldsymbol{x}_i, \tilde{\boldsymbol{w}}_m\right))| \\
&\qquad\qquad + |(\alpha(\boldsymbol{w}_{m,n}^\top \boldsymbol{x}_i - \tilde{\boldsymbol{w}}_{m,n}^\top \boldsymbol{x}_i))| \Big) \\
&\leq 2\alpha \|\boldsymbol{x}_i\|_2^2 \sum_{n=1}^{\mathcal{N}} \|\boldsymbol{w}_{m,n} - \tilde{\boldsymbol{w}}_{m,n}\|_2. \qquad\qquad \text{(D.20)}
\end{aligned}
$$

To link Equation (D.19) and Equation (D.20) to the $\|\boldsymbol{\theta} - \tilde{\boldsymbol{\theta}}\|_2$, we use Lemma 7 to obtain the following inequalities:

$$
\sum_{n=1}^{\mathcal{N}} \|\boldsymbol{w}_{m,n} - \tilde{\boldsymbol{w}}_{m,n}\|_2 \leq \sqrt{\mathcal{N}} \|\boldsymbol{w}_m - \tilde{\boldsymbol{w}}_m\|_2 \leq \sqrt{\mathcal{N}} \|\boldsymbol{\theta} - \tilde{\boldsymbol{\theta}}\|_2, \qquad\qquad \text{(D.21)}
$$

$$
\sum_{\ell=1}^{\mathcal{L}} |\pi_{m,\ell} - \tilde{\pi}_{m,\ell}| \leq \sqrt{\mathcal{L}} \|\boldsymbol{\pi}_m - \tilde{\boldsymbol{\pi}}_m\|_2 \leq \sqrt{\mathcal{L}} \|\boldsymbol{\theta} - \tilde{\boldsymbol{\theta}}\|_2. \qquad\qquad \text{(D.22)}
$$

With Equation (D.1), Equation (D.16), Equation (D.19), Equation (D.20), Equation (D.21), and Equation (D.22),

$$
\begin{aligned}
&\|\boldsymbol{J}(\boldsymbol{x}, \boldsymbol{\theta}) - \boldsymbol{J}(\boldsymbol{x}, \tilde{\boldsymbol{\theta}})\|_F^2 \\
&= \sum_{i=1}^{N} (\|\boldsymbol{J}(\boldsymbol{x}_i, \boldsymbol{w}) - \boldsymbol{J}(\boldsymbol{x}_i, \tilde{\boldsymbol{w}})\|_F^2 + \|\boldsymbol{J}(\boldsymbol{x}_i, \boldsymbol{\pi}) - \boldsymbol{J}(\boldsymbol{x}_i, \tilde{\boldsymbol{\pi}})\|_F^2) \\
&= \frac{1}{M} \sum_{i=1}^{N} \sum_{m=1}^{M} \left( \sum_{n=1}^{\mathcal{N}} \left( \left\| \sum_{\ell=1}^{\mathcal{L}} \left( \pi_{m,\ell} \frac{\partial \mu_{m,\ell}(\boldsymbol{x}_i, \boldsymbol{w}_m)}{\partial \boldsymbol{w}_{m,n}} - \tilde{\pi}_{m,\ell} \frac{\partial \mu_{m,\ell}(\boldsymbol{x}_i, \tilde{\boldsymbol{w}}_m)}{\partial \tilde{\boldsymbol{w}}_{m,n}} \right) \right\|_2^2 \right) \right. \\
&\qquad\qquad\qquad \left. + \sum_{\ell=1}^{\mathcal{L}} (\mu_{m,\ell}(\boldsymbol{x}_i, \boldsymbol{w}_m) - \mu_{m,\ell}(\boldsymbol{x}_i, \tilde{\boldsymbol{w}}_m))^2 \right) \\
&= \frac{1}{M} \sum_{i=1}^{N} \sum_{m=1}^{M} \left( \sum_{n=1}^{\mathcal{N}} \left( \left\| \sum_{\ell=1}^{\mathcal{L}} \left( (\pi_{m,\ell} - \tilde{\pi}_{m,\ell}) \frac{\partial \mu_{m,\ell}(\boldsymbol{x}_i, \boldsymbol{w}_m)}{\partial \boldsymbol{w}_{m,n}} + \left( \frac{\partial \mu_{m,\ell}(\boldsymbol{x}_i, \boldsymbol{w}_m)}{\partial \boldsymbol{w}_{m,n}} - \frac{\partial \mu_{m,\ell}(\boldsymbol{x}_i, \tilde{\boldsymbol{w}}_m)}{\partial \tilde{\boldsymbol{w}}_{m,n}} \right) \tilde{\pi}_{m,\ell} \right) \right\|_2^2 \right) \right. \\
&\qquad\qquad\qquad \left. + \sum_{\ell=1}^{\mathcal{L}} (\mu_{m,\ell}(\boldsymbol{x}_i, \boldsymbol{w}_m) - \mu_{m,\ell}(\boldsymbol{x}_i, \tilde{\boldsymbol{w}}_m))^2 \right)
\end{aligned}
$$

$$
\leq \frac{1}{M} \sum_{i=1}^{N} \sum_{m=1}^{M} \left( \sum_{n=1}^{\mathcal{N}} \left( \left( \sum_{\ell=1}^{\mathcal{L}} (|\pi_{m,\ell} - \tilde{\pi}_{m,\ell}|\alpha\|\boldsymbol{x}_i\|_2) + \left( 2\alpha\|\boldsymbol{x}_i\|_2^2 \sum_{n=1}^{\mathcal{N}} \|\boldsymbol{w}_{m,n} - \tilde{\boldsymbol{w}}_{m,n}\|_2 v\mathcal{L} \right) \right)^2 \right) \right.
$$

$$
\left. + \sum_{\ell=1}^{\mathcal{L}} \left( \sum_{n=1}^{\mathcal{N}} \|\boldsymbol{x}_i\|_2 \|\boldsymbol{w}_{m,n} - \tilde{\boldsymbol{w}}_{m,n}\|_2 \right)^2 \right)
$$

$$
\leq \frac{1}{M} \sum_{i=1}^{N} \sum_{m=1}^{M} \left( \sum_{n=1}^{\mathcal{N}} \left( \left( \left( \sqrt{\mathcal{L}}\|\boldsymbol{\theta} - \tilde{\boldsymbol{\theta}}\|_2 \alpha\|\boldsymbol{x}_i\|_2 \right) + \left( 2\alpha\|\boldsymbol{x}_i\|_2^2 \sqrt{\mathcal{N}}\|\boldsymbol{\theta} - \tilde{\boldsymbol{\theta}}\|_2 v\mathcal{L} \right) \right)^2 \right) \right.
$$

$$
\left. + \sum_{\ell=1}^{\mathcal{L}} \left( \sqrt{\mathcal{N}}\|\boldsymbol{x}_i\|_2\|\boldsymbol{\theta} - \tilde{\boldsymbol{\theta}}\|_2 \right)^2 \right)
$$

$$
\leq \sum_{i=1}^{N} \left( \mathcal{N}\left( \alpha\sqrt{\mathcal{L}}\|\boldsymbol{x}_i\|_2 + 2\alpha\|\boldsymbol{x}_i\|_2^2 \sqrt{\mathcal{N}}v\mathcal{L} \right)^2 + \mathcal{L}\mathcal{N}\|\boldsymbol{x}_i\|_2^2 \right) \|\boldsymbol{\theta} - \tilde{\boldsymbol{\theta}}\|_2^2. \tag{D.23}
$$

By considering the square root of both sides in Equation (D.17) and Equation (D.23), we conclude the proof for Lemma 8. $\square$

## E  PROOF OF THEOREM 3

*Proof.* We can use the same approach with the proof of Theorem 1. Using an incremental formula, the output from the oblivious tree ensembles can be written as follows:

$$
f^{(d)}(\boldsymbol{x}_i, \boldsymbol{w}, \boldsymbol{\pi}) = \frac{1}{\sqrt{M}} \sum_{m=1}^{M} \left( \sigma\left(\boldsymbol{w}_{m,t}^{\top}\boldsymbol{x}_i\right) f_m^{(d-1)}\left(\boldsymbol{x}_i, \boldsymbol{w}_m^{(s)}, \boldsymbol{\pi}_m^{(l)}\right) \right.
$$

$$
\left. + \left(1 - \sigma\left(\boldsymbol{w}_{m,t}^{\top}\boldsymbol{x}_i\right)\right) f_m^{(d-1)}\left(\boldsymbol{x}_i, \boldsymbol{w}_m^{(s)}, \boldsymbol{\pi}_m^{(r)}\right) \right), \tag{E.1}
$$

where $(s)$ of $\boldsymbol{w}_m^{(s)}$ means (s)hared parameters at subtrees. Intuitively, the fundamental of Theorem 3 is that the outputs of the left subtree and right subtree are still independent with the oblivious tree structure. Even with parameter sharing at the same depth, since the leaf parameters $\boldsymbol{\pi}$ are not shared, the outputs of the left subtree and right subtree are independent.

We will see that Lemma 1, 2 and 3 are also valid for oblivious tree ensembles.

**Correspondence to Lemma 1.** To show the correspondence to Lemma 1, it is sufficient to show that Equation (A.22), Equation (A.23), Equation (A.24), and Equation (A.25) hold when

$$
\mathbb{E}_m \left[ f_m^{(d+1)}(\boldsymbol{x}_i, \boldsymbol{w}_m, \boldsymbol{\pi}_m) f_m^{(d+1)}\left(\boldsymbol{x}_j, \boldsymbol{w}_m, \boldsymbol{\pi}_m\right) \right]
$$

$$
= \mathbb{E}_m \left[ \left( \underbrace{\left( f_m^{(d)}\left(\boldsymbol{x}_i, \boldsymbol{w}_m^{(s)}, \boldsymbol{\pi}_m^{(l)}\right) - f_m^{(d)}\left(\boldsymbol{x}_i, \boldsymbol{w}_m^{(s)}, \boldsymbol{\pi}_m^{(r)}\right) \right)\sigma(\boldsymbol{w}_{m,t}^{\top}\boldsymbol{x}_i)}_{(A)} + \underbrace{f_m^{(d)}\left(\boldsymbol{x}_i, \boldsymbol{w}_m^{(s)}, \boldsymbol{\pi}_m^{(r)}\right)}_{(B)} \right)
$$

$$
\left( \underbrace{\left( f_m^{(d)}\left(\boldsymbol{x}_j, \boldsymbol{w}_m^{(s)}, \boldsymbol{\pi}_m^{(l)}\right) - f_m^{(d)}\left(\boldsymbol{x}_j, \boldsymbol{w}_m^{(s)}, \boldsymbol{\pi}_m^{(r)}\right) \right)\sigma(\boldsymbol{w}_{m,t}^{\top}\boldsymbol{x}_j)}_{(C)} + \underbrace{f_m^{(d)}\left(\boldsymbol{x}_j, \boldsymbol{w}_m^{(s)}, \boldsymbol{\pi}_m^{(r)}\right)}_{(D)} \right) \right].
$$
$$
\tag{E.2}
$$

This equation corresponds to Equation (A.21). Here, since the leaf parameters $\boldsymbol{\pi}$ are not shared, the outputs of the left subtree and right subtree are still independent even with the oblivious tree structure. Therefore, we can obtain the correspondences to Equation (A.22), Equation (A.23), Equation (A.24), and Equation (A.25) with the same procedures.

**Correspondence to Lemma 2.** For the depth $d + 1$, since

$$\frac{\partial f^{(d+1)}(\boldsymbol{x}_i, \boldsymbol{w}, \boldsymbol{\pi})}{\partial \boldsymbol{w}_{m,s}} = \sigma(\boldsymbol{w}_{m,t}^\top \boldsymbol{x}_i) \frac{\partial f_m^{(d)}\left(\boldsymbol{x}_i, \boldsymbol{w}_m^{(s)}, \boldsymbol{\pi}_m^{(r)}\right)}{\partial \boldsymbol{w}_{m,s}}$$

$$+ \left(1 - \sigma(\boldsymbol{w}_{m,t}^\top \boldsymbol{x}_i)\right) \frac{\partial f_m^{(d)}\left(\boldsymbol{x}_i, \boldsymbol{w}_m^{(s)}, \boldsymbol{\pi}_m^{(r)}\right)}{\partial \boldsymbol{w}_{m,s}}, \qquad \text{(E.3)}$$

the corresponding limiting TNTK is

$$\Theta^{(d+1),(s)}(\boldsymbol{x}_i, \boldsymbol{x}_j)$$

$$= \sum_{s=2}^d \mathbb{E}_m \left[ \left( \sigma(\boldsymbol{w}_{m,t}^\top \boldsymbol{x}_i) \frac{\partial f_m^{(d)}\left(\boldsymbol{x}_i, \boldsymbol{w}_m^{(s)}, \boldsymbol{\pi}_m^{(r)}\right)}{\partial \boldsymbol{w}_{m,s}} + \left(1 - \sigma(\boldsymbol{w}_{m,t}^\top \boldsymbol{x}_i)\right) \frac{\partial f_m^{(d)}\left(\boldsymbol{x}_i, \boldsymbol{w}_m^{(s)}, \boldsymbol{\pi}_m^{(r)}\right)}{\partial \boldsymbol{w}_{m,s}} \right)^\top \right.$$

$$\left. \left( \sigma(\boldsymbol{w}_{m,t}^\top \boldsymbol{x}_j) \frac{\partial f_m^{(d)}\left(\boldsymbol{x}_j, \boldsymbol{w}_m^{(s)}, \boldsymbol{\pi}_m^{(r)}\right)}{\partial \boldsymbol{w}_{m,s}} + \left(1 - \sigma(\boldsymbol{w}_{m,t}^\top \boldsymbol{x}_j)\right) \frac{\partial f_m^{(d)}\left(\boldsymbol{x}_j, \boldsymbol{w}_m^{(s)}, \boldsymbol{\pi}_m^{(r)}\right)}{\partial \boldsymbol{w}_{m,s}} \right) \right]$$

$$= \sum_{s=2}^d \mathbb{E}_m \left[ \left( \underbrace{\sigma(\boldsymbol{w}_{m,t}^\top \boldsymbol{x}_i) \left( \frac{\partial f_m^{(d)}\left(\boldsymbol{x}_i, \boldsymbol{w}_m^{(s)}, \boldsymbol{\pi}_m^{(l)}\right)}{\partial \boldsymbol{w}_{m,s}} - \frac{\partial f_m^{(d)}\left(\boldsymbol{x}_i, \boldsymbol{w}_m^{(s)}, \boldsymbol{\pi}_m^{(r)}\right)}{\partial \boldsymbol{w}_{m,s}} \right)}_{\text{(A)}} + \underbrace{\frac{\partial f_m^{(d)}\left(\boldsymbol{x}_i, \boldsymbol{w}_m^{(s)}, \boldsymbol{\pi}_m^{(r)}\right)}{\partial \boldsymbol{w}_{m,s}}}_{\text{(B)}} \right)^\top \right.$$

$$\left. \left( \underbrace{\sigma(\boldsymbol{w}_{m,t}^\top \boldsymbol{x}_j) \left( \frac{\partial f_m^{(d)}\left(\boldsymbol{x}_j, \boldsymbol{w}_m^{(s)}, \boldsymbol{\pi}_m^{(l)}\right)}{\partial \boldsymbol{w}_{m,s}} - \frac{\partial f_m^{(d)}\left(\boldsymbol{x}_j, \boldsymbol{w}_m^{(s)}, \boldsymbol{\pi}_m^{(r)}\right)}{\partial \boldsymbol{w}_{m,s}} \right)}_{\text{(C)}} + \underbrace{\frac{\partial f_m^{(d)}\left(\boldsymbol{x}_j, \boldsymbol{w}_m^{(s)}, \boldsymbol{\pi}_m^{(r)}\right)}{\partial \boldsymbol{w}_{m,s}}}_{\text{(D)}} \right) \right].$$

$$\text{(E.4)}$$

Since $\frac{\partial f_m^{(d)}\left(\boldsymbol{x}_i, \boldsymbol{w}_m^{(s)}, \boldsymbol{\pi}_m^{(r)}\right)}{\partial \boldsymbol{w}_{m,s}}$ and $\frac{\partial f_m^{(d)}\left(\boldsymbol{x}_j, \boldsymbol{w}_m^{(s)}, \boldsymbol{\pi}_m^{(l)}\right)}{\partial \boldsymbol{w}_{m,s}}$ for $s = \{2, 3, \ldots, d\}$ are independent to each other and have zero-mean Gaussian distribution[4], similar calculation used for Equation (A.22), Equa-

---

[4]For a single oblivious tree, the number splitting rule is $d$ because of the parameter sharing.

tion (A.23), Equation (A.24), and Equation (A.25) gives

$$
\mathbb{E}_m\left[(A) \times (C)\right] = \mathcal{T}\left(\boldsymbol{x}_i, \boldsymbol{x}_j\right) \mathbb{E}_m\left[\left(\frac{\partial f_m^{(d)}\left(\boldsymbol{x}_i, \boldsymbol{w}_m^{(s)}, \boldsymbol{\pi}_m^{(l)}\right)}{\partial \boldsymbol{w}_{m,s}}\right)^\top \left(\frac{\partial f_m^{(d)}\left(\boldsymbol{x}_j, \boldsymbol{w}_m^{(s)}, \boldsymbol{\pi}_m^{(l)}\right)}{\partial \boldsymbol{w}_{m,s}}\right)\right.
$$
$$
\left. + \left(\frac{\partial f_m^{(d)}\left(\boldsymbol{x}_i, \boldsymbol{w}_m^{(s)}, \boldsymbol{\pi}_m^{(r)}\right)}{\partial \boldsymbol{w}_{m,s}}\right)^\top \left(\frac{\partial f_m^{(d)}\left(\boldsymbol{x}_j, \boldsymbol{w}_m^{(s)}, \boldsymbol{\pi}_m^{(r)}\right)}{\partial \boldsymbol{w}_{m,s}}\right)\right],
$$
$$(E.5)$$

$$
\mathbb{E}_m\left[(B) \times (C)\right] = -0.5\, \mathbb{E}_m\left[\left(\frac{\partial f_m^{(d)}\left(\boldsymbol{x}_i, \boldsymbol{w}_m^{(s)}, \boldsymbol{\pi}_m^{(r)}\right)}{\partial \boldsymbol{w}_{m,s}}\right)^\top \left(\frac{\partial f_m^{(d)}\left(\boldsymbol{x}_j, \boldsymbol{w}_m^{(s)}, \boldsymbol{\pi}_m^{(r)}\right)}{\partial \boldsymbol{w}_{m,s}}\right)\right], \quad (E.6)
$$

$$
\mathbb{E}_m\left[(A) \times (D)\right] = -0.5\, \mathbb{E}_m\left[\left(\frac{\partial f_m^{(d)}\left(\boldsymbol{x}_i, \boldsymbol{w}_m^{(s)}, \boldsymbol{\pi}_m^{(r)}\right)}{\partial \boldsymbol{w}_{m,s}}\right)^\top \left(\frac{\partial f_m^{(d)}\left(\boldsymbol{x}_j, \boldsymbol{w}_m^{(s)}, \boldsymbol{\pi}_m^{(r)}\right)}{\partial \boldsymbol{w}_{m,s}}\right)\right], \quad (E.7)
$$

$$
\mathbb{E}_m\left[(B) \times (D)\right] = \mathbb{E}_m\left[\left(\frac{\partial f_m^{(d)}\left(\boldsymbol{x}_i, \boldsymbol{w}_m^{(s)}, \boldsymbol{\pi}_m^{(r)}\right)}{\partial \boldsymbol{w}_{m,s}}\right)^\top \left(\frac{\partial f_m^{(d)}\left(\boldsymbol{x}_j, \boldsymbol{w}_m^{(s)}, \boldsymbol{\pi}_m^{(r)}\right)}{\partial \boldsymbol{w}_{m,s}}\right)\right]. \quad (E.8)
$$

As in the previous calculations, Equation (E.6), Equation (E.7), and Equation (E.8) cancel each other out. As a result, we obtain

$$
\Theta^{(d+1),(s)}\left(\boldsymbol{x}_i, \boldsymbol{x}_j\right) = 2\mathcal{T}\left(\boldsymbol{x}_i, \boldsymbol{x}_j\right)\left(\Theta^{(d),(t)}\left(\boldsymbol{x}_i, \boldsymbol{x}_j\right) + \Theta^{(d),(s)}\left(\boldsymbol{x}_i, \boldsymbol{x}_j\right)\right). \quad (E.9)
$$

**Correspondence to Lemma 3.** Considering Equation (A.33), once we focus on a leaf $\ell$, it is not possible for both $\mathbb{1}_{1\searrow\ell}$ and $\mathbb{1}_{\ell\swarrow 1}$ to be 1. This means that a leaf cannot belong to both the right subtree and the left subtree. Therefore, even with the oblivious tree structure, there are no influences. Therefore, we get exactly the same result for the Lemma 3. $\qquad\square$

# F  DETAILS OF NUMERICAL EXPERIMENTS

## F.1  SETUP

### F.1.1  DATASET ACQUISITION

We use the UCI datasets (Dua & Graff, 2017) preprocessed by Fernández-Delgado et al. (2014), which are publicly available at http://persoal.citius.usc.es/manuel.fernandez. delgado/papers/jmlr/data.tar.gz. Since the size of the kernel is the square of the dataset size and too many data make training impractical, we use preprocessed UCI datasets with the number of samples smaller than 5000. Arora et al. (2020) reported the bug in the preprocess when the explicit training/test split is given. Therefore, we do not use that dataset with explicit training/test split. As a consequence, 90 different datasets are available.

### F.1.2  KERNEL SPECIFICATIONS

**TNTK.** See Theorem 1 for the detailed definitions. We change the tree depth from 1 to 29 and change $\alpha$ in $\{0.5, 1.0, 2.0, 4.0, 8.0, 16.0, 32.0, 64.0\}$.

**MLP-induced NTK.** We assume the MLP activation function as ReLU. Our implementation is based on the publicly available code[5] used in Arora et al. (2020). For detailed definitions, see Arora et al. (2020). The hyperparameter of this kernel is the model depth. We change the depth from 1 to 29. Here, depth $= 1$ means there is no hidden layer in the MLP.

---

[5]https://github.com/LeoYu/neural-tangent-kernel-UCI

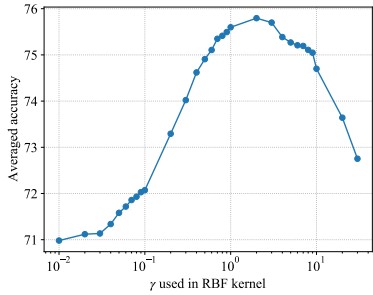

Figure 9: The $\gamma$ dependency of the RBF kernel performance.

**RBF kernel.** We use scikit-learn implementation[6]. The hyperparameter of this kernel is $\gamma$, inverse of the standard deviation of the RBF kernel (Gaussian function). For Figure 3, we tune $\gamma$ in $\{0.01, 0.02, 0.03, 0.04, 0.05, 0.06, 0.07, 0.08, 0.09, 0.1, 0.2, 0.3, 0.4, 0.5, 0.6, 0.7, 0.8, 0.9, 1.0, 2.0, 3.0, 4.0, 5.0, 6.0, 7.0, 8.0, 9.0, 10.0, 20.0, 30.0\}$, resulting in 30 candidates in total. In our experiments, $\gamma = 2.0$ performs the best on average (Figure 9).

### F.1.3 MODEL SPECIFICATIONS

We used kernel regression implemented in scikit-learn[7]. To consider ridge-less situation, regularization strength is set to be $1.0 \times 10^{-8}$, a very small constant.

### F.1.4 COMPUTATIONAL COSTS

Since the training and inference algorithms of the kernel regression are common across different kernels, we analyze the computational cost of computing a single value in a gram matrix of the corresponding kernels in the following. The time complexity of the MLP-induced NTK is linear with respect to the layer depth, while that of the TNTK remains to be constant. Such a trend can be seen in the right panel of Figure 7. For the RBF kernel, the computational cost remains the same with respect to changes in hyperparameters, thus its trend is similar to the TNTK. In terms of the space complexity, when considering a multi-layered MLP, since it is not necessary to store all past calculation results in memory during the recursive computation, the MLP-induced NTK computation consumes a certain amount of memory regardless of the depth of the layers. Therefore, the memory usage is almost the same across the RBF kernel, TNTK, and MLP-induced NTK.

### F.1.5 COMPUTATIONAL RESOURCE

We used Ubuntu Linux (version: 4.15.0-117-generic) and ran all experiments on 2.20 GHz Intel Xeon E5-2698 CPU and 252 GB of memory.

### F.2 RESULTS

### F.2.1 STATISTICAL SIGNIFICANCE OF THE PARAMETER DEPENDENCY

A Wilcoxon signed rank test is conducted to check the statistical significance of the differences between different $\alpha$. Figure 10 shows the p-values for the depth of 3 and 20. As shown in Figure 7, when the tree is shallow, the accuracy started to deteriorate after around $\alpha = 8.0$, but as the tree becomes deeper, the deterioration became less apparent. Therefore, statistically significantly different pairs for deep tress and shallow trees are different. When the tree is deep, large $\alpha$ shows a significant difference over those with small $\alpha$. However, when the tree is shallow, the best performance is achieved with $\alpha$ of about 8.0, and if $\alpha$ is too large, the performance deteriorates predominantly.

---

[6]https://scikit-learn.org/stable/modules/generated/sklearn.metrics.pairwise.rbf_kernel.html
[7]https://scikit-learn.org/stable/modules/generated/sklearn.kernel_ridge.KernelRidge.html

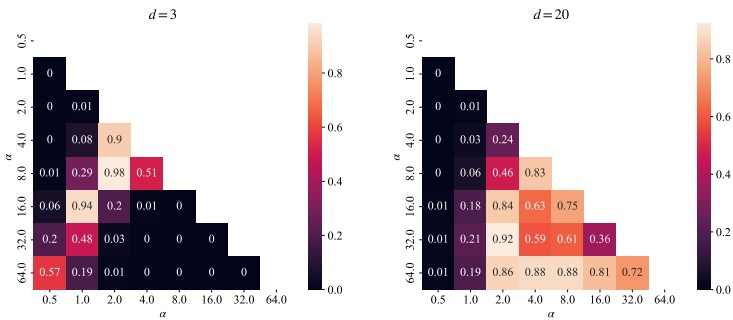

Figure 10: P-values of the Wilcoxon signed rank test for different pairs of $\alpha$.

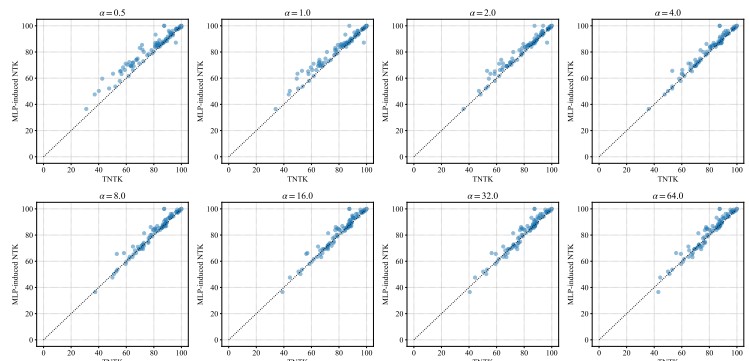

Figure 11: Performance comparisons between the kernel regression with MLP-induced NTK and the TNTK on the UCI dataset.

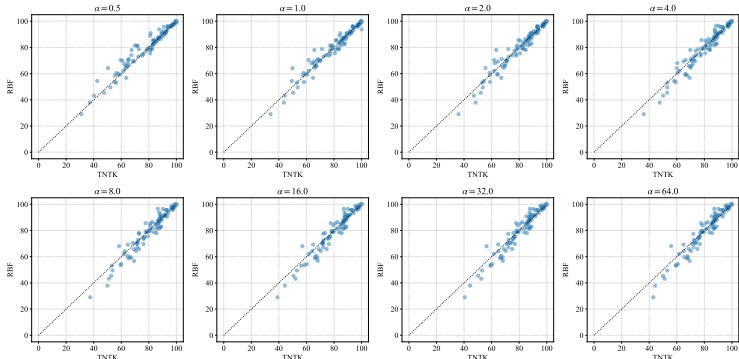

Figure 12: Performance comparisons between the kernel regression with RBF kernel and the TNTK on the UCI dataset.

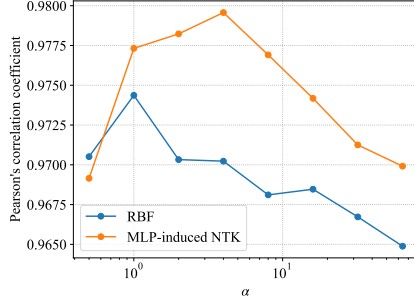

Figure 13: Pearson's correlation coefficients with predicted values of the TNTK with different $\alpha$.

### F.2.2 DATASET-WISE RESULTS

For each $\alpha$, scatter-plots are shown in Figures 11 and 12. As shown in Figure 13, the correlation coefficients with the TNTK are likely to be higher for the MLP-induced NTK than for the RBF kernel. Tables 2, 3 and 4 are dataset-wise results of the comparison between the TNTK, the MLP-induced NTK, and the RBF kernel. For each $\alpha$, depth is tuned for each dataset. In terms of the depth, the best performers from 1 to 29 are compared with the TNTK and the MLP-induced NTK. For the RBF kernel, $\gamma$ is tuned in each dataset from 30 candidate values as described in Section F.1.2. Therefore, the number of tunable parameters is the same across all methods. All parameter-wise results are visualized in Figures 14 and 15.

| | name | size | α=0.5 | α=1.0 | α=2.0 | α=4.0 | α=8.0 | α=16.0 | α=32.0 | α=64.0 | MLP-NTK | RBF |
|---|---|---|---|---|---|---|---|---|---|---|---|---|
| 0 | trains | 10 | 87.500 | 87.500 | 87.500 | 87.500 | 87.500 | 87.500 | 87.500 | 87.500 | 100.000 | 87.500 |
| 1 | balloons | 16 | 87.500 | 100.000 | 93.750 | 87.500 | 87.500 | 87.500 | 87.500 | 87.500 | 100.000 | 93.750 |
| 2 | lenses | 24 | 87.500 | 87.500 | 87.500 | 87.500 | 87.500 | 87.500 | 87.500 | 87.500 | 87.500 | 87.500 |
| 3 | lung-cancer | 32 | 56.250 | 53.125 | 53.125 | 53.125 | 53.125 | 56.250 | 59.375 | 59.375 | 65.625 | 53.125 |
| 4 | post-operative | 90 | 63.636 | 64.773 | 67.045 | 69.318 | 69.318 | 68.182 | 68.182 | 69.318 | 69.318 | 56.818 |
| 5 | pittsburg-bridges-SPAN | 92 | 55.435 | 57.609 | 58.696 | 67.391 | 65.217 | 66.304 | 66.304 | 67.391 | 65.217 | 58.696 |
| 6 | fertility | 100 | 84.000 | 88.000 | 89.000 | 89.000 | 89.000 | 89.000 | 89.000 | 89.000 | 89.000 | 83.000 |
| 7 | zoo | 101 | 100.000 | 99.000 | 99.000 | 99.000 | 99.000 | 99.000 | 99.000 | 99.000 | 99.000 | 99.000 |
| 8 | pittsburg-bridges-T-OR-D | 102 | 81.000 | 84.000 | 87.000 | 89.000 | 89.000 | 89.000 | 88.000 | 88.000 | 87.000 | 89.000 |
| 9 | pittsburg-bridges-REL-L | 103 | 67.308 | 74.038 | 75.000 | 74.038 | 75.962 | 75.962 | 75.962 | 75.962 | 74.038 | 74.038 |
| 10 | pittsburg-bridges-TYPE | 105 | 57.692 | 59.615 | 64.423 | 66.346 | 66.346 | 66.346 | 66.346 | 65.385 | 68.269 | 59.615 |
| 11 | molec-biol-promoter | 106 | 90.385 | 88.462 | 87.500 | 87.500 | 87.500 | 87.500 | 87.500 | 87.500 | 90.385 | 88.462 |
| 12 | pittsburg-bridges-MATERIAL | 106 | 93.269 | 94.231 | 94.231 | 94.231 | 94.231 | 94.231 | 94.231 | 94.231 | 94.231 | 93.269 |
| 13 | breast-tissue | 106 | 65.385 | 68.269 | 67.308 | 70.192 | 72.115 | 72.115 | 75.000 | 74.038 | 69.231 | 71.154 |
| 14 | acute-nephritis | 120 | 100.000 | 100.000 | 100.000 | 100.000 | 100.000 | 100.000 | 100.000 | 100.000 | 100.000 | 100.000 |
| 15 | acute-inflammation | 120 | 100.000 | 100.000 | 100.000 | 100.000 | 100.000 | 100.000 | 100.000 | 100.000 | 100.000 | 100.000 |
| 16 | heart-switzerland | 123 | 37.097 | 43.548 | 48.387 | 47.581 | 50.000 | 44.355 | 44.355 | 44.355 | 47.581 | 37.903 |
| 17 | echocardiogram | 131 | 81.061 | 81.061 | 84.848 | 84.091 | 85.606 | 85.606 | 85.606 | 85.606 | 85.606 | 78.030 |
| 18 | lymphography | 148 | 88.514 | 88.514 | 88.514 | 88.514 | 87.838 | 87.162 | 86.486 | 86.486 | 88.514 | 86.486 |
| 19 | iris | 150 | 95.946 | 97.973 | 96.622 | 88.514 | 86.486 | 87.162 | 87.838 | 87.162 | 87.162 | 96.622 |
| 20 | teaching | 151 | 56.579 | 57.895 | 58.553 | 60.526 | 64.474 | 67.105 | 67.105 | 67.763 | 63.158 | 60.526 |
| 21 | hepatitis | 155 | 83.974 | 85.256 | 85.256 | 84.615 | 84.615 | 84.615 | 84.615 | 85.256 | 83.974 | 85.256 |
| 22 | wine | 178 | 98.864 | 99.432 | 98.864 | 98.864 | 98.864 | 98.295 | 98.295 | 97.727 | 99.432 | 98.295 |
| 23 | planning | 182 | 62.222 | 65.556 | 70.000 | 71.667 | 71.667 | 72.222 | 72.222 | 72.222 | 72.222 | 67.222 |
| 24 | flags | 194 | 52.083 | 53.646 | 53.646 | 53.125 | 53.646 | 53.125 | 53.125 | 53.125 | 53.646 | 49.479 |
| 25 | parkinsons | 195 | 93.878 | 94.388 | 92.857 | 93.367 | 92.857 | 92.857 | 93.367 | 93.367 | 93.878 | 95.408 |
| 26 | breast-cancer-wisc-prog | 198 | 82.143 | 83.673 | 83.673 | 83.673 | 83.673 | 83.673 | 83.673 | 83.673 | 85.204 | 78.571 |
| 27 | heart-va | 200 | 31.000 | 34.000 | 36.000 | 36.000 | 37.500 | 39.000 | 40.500 | 43.000 | 36.500 | 29.000 |
| 28 | conn-bench-sonar-mines-rocks | 208 | 86.538 | 86.538 | 87.019 | 86.538 | 86.538 | 86.538 | 86.538 | 86.538 | 87.981 | 87.500 |
| 29 | seeds | 210 | 90.865 | 93.750 | 93.269 | 91.827 | 92.308 | 92.308 | 91.827 | 91.827 | 96.154 | 95.673 |

Table 2: Comparison between TNTK and MLP-induced NTK for a half of the dataset (1/3).

| | name | size | α=0.5 | α=1.0 | α=2.0 | α=4.0 | α=8.0 | α=16.0 | α=32.0 | α=64.0 | MLP-NTK | RBF |
|---|---|---|---|---|---|---|---|---|---|---|---|---|
| 30 | glass | 214 | 60.849 | 67.453 | 70.755 | 70.755 | 71.698 | 71.698 | 71.226 | 71.226 | 70.283 | 69.811 |
| 31 | statlog-heart | 270 | 83.209 | 87.313 | 87.687 | 88.433 | 88.433 | 88.433 | 87.687 | 87.313 | 86.567 | 82.463 |
| 32 | breast-cancer | 286 | 64.789 | 67.254 | 69.718 | 70.423 | 72.887 | 74.648 | 75.000 | 75.000 | 71.831 | 65.845 |
| 33 | heart-hungarian | 294 | 83.219 | 83.904 | 84.247 | 84.589 | 85.616 | 85.274 | 85.274 | 85.274 | 85.616 | 82.877 |
| 34 | heart-cleveland | 303 | 55.263 | 58.224 | 58.882 | 59.211 | 59.539 | 58.553 | 59.211 | 59.539 | 57.895 | 53.618 |
| 35 | haberman-survival | 306 | 59.539 | 61.513 | 61.842 | 66.447 | 68.421 | 71.711 | 73.684 | 73.684 | 71.053 | 70.395 |
| 36 | vertebral-column-2clases | 310 | 69.805 | 77.273 | 78.571 | 80.844 | 82.792 | 84.091 | 84.091 | 82.792 | 83.117 | 81.494 |
| 37 | vertebral-column-3clases | 310 | 71.753 | 78.247 | 81.169 | 80.844 | 80.844 | 81.818 | 81.494 | 81.169 | 81.818 | 81.169 |
| 38 | primary-tumor | 330 | 47.561 | 50.305 | 52.134 | 53.049 | 52.744 | 50.610 | 50.305 | 50.000 | 52.134 | 45.427 |
| 39 | ecoli | 336 | 71.429 | 79.167 | 83.036 | 84.524 | 86.012 | 86.607 | 86.905 | 86.905 | 85.417 | 81.250 |
| 40 | ionosphere | 351 | 90.057 | 91.477 | 90.341 | 87.784 | 88.352 | 88.352 | 88.352 | 88.352 | 91.761 | 92.330 |
| 41 | libras | 360 | 82.778 | 81.389 | 80.556 | 80.833 | 81.111 | 80.833 | 80.833 | 80.833 | 83.889 | 85.278 |
| 42 | dermatology | 366 | 97.802 | 97.802 | 97.527 | 97.527 | 97.253 | 97.253 | 97.253 | 97.253 | 97.802 | 97.253 |
| 43 | congressional-voting | 435 | 61.697 | 61.697 | 61.927 | 61.927 | 61.927 | 61.697 | 61.697 | 61.697 | 61.697 | 62.156 |
| 44 | arrhythmia | 452 | 69.469 | 65.265 | 64.602 | 64.823 | 64.823 | 64.823 | 64.823 | 64.823 | 71.239 | 69.248 |
| 45 | musk-1 | 476 | 89.076 | 89.076 | 89.076 | 89.286 | 89.286 | 89.076 | 89.076 | 89.076 | 89.706 | 90.756 |
| 46 | cylinder-bands | 512 | 79.883 | 78.125 | 78.125 | 78.320 | 78.516 | 78.320 | 78.320 | 78.320 | 80.273 | 79.688 |
| 47 | low-res-spect | 531 | 91.729 | 91.353 | 90.602 | 89.474 | 88.534 | 87.782 | 87.218 | 87.218 | 91.353 | 90.226 |
| 48 | breast-cancer-wisc-diag | 569 | 96.127 | 96.655 | 97.359 | 97.359 | 97.359 | 96.831 | 96.479 | 96.479 | 97.007 | 95.599 |
| 49 | ilpd-indian-liver | 583 | 64.897 | 69.521 | 70.719 | 72.260 | 71.062 | 71.747 | 72.603 | 72.603 | 71.918 | 70.377 |
| 50 | synthetic-control | 600 | 99.333 | 99.333 | 99.167 | 98.833 | 98.333 | 97.833 | 97.000 | 96.667 | 98.833 | 99.333 |
| 51 | balance-scale | 625 | 81.250 | 84.615 | 88.782 | 89.904 | 91.346 | 90.064 | 85.256 | 85.256 | 93.269 | 90.865 |
| 52 | statlog-australian-credit | 690 | 59.012 | 60.610 | 64.099 | 66.279 | 67.151 | 68.023 | 68.023 | 68.023 | 66.279 | 59.302 |
| 53 | credit-approval | 690 | 82.558 | 85.174 | 86.628 | 87.209 | 87.645 | 87.791 | 87.791 | 87.355 | 87.064 | 81.686 |
| 54 | breast-cancer-wisc | 699 | 96.286 | 97.286 | 97.857 | 97.857 | 98.000 | 98.000 | 98.000 | 98.000 | 98.000 | 96.714 |
| 55 | blood | 748 | 67.513 | 65.775 | 63.369 | 69.786 | 72.727 | 73.529 | 75.802 | 77.005 | 74.064 | 78.075 |
| 56 | energy-y2 | 768 | 89.583 | 89.453 | 88.021 | 87.891 | 88.151 | 87.630 | 87.630 | 87.630 | 88.281 | 90.755 |
| 57 | pima | 768 | 68.229 | 70.182 | 71.354 | 73.307 | 76.042 | 76.302 | 77.083 | 76.693 | 75.000 | 69.661 |
| 58 | energy-y1 | 768 | 93.750 | 93.620 | 93.229 | 90.495 | 90.234 | 90.104 | 90.234 | 90.234 | 92.708 | 96.484 |
| 59 | statlog-vehicle | 846 | 78.318 | 77.014 | 76.540 | 73.578 | 72.986 | 72.156 | 72.038 | 72.038 | 81.398 | 77.488 |

Table 3: Comparison between TNTK and MLP-induced NTK for a half of the dataset (2/3).

| | name | size | α=0.5 | α=1.0 | α=2.0 | α=4.0 | α=8.0 | α=16.0 | α=32.0 | α=64.0 | MLP-NTK | RBF |
|---|---|---|---|---|---|---|---|---|---|---|---|---|
| 60 | oocytes_trisopterus_nucleus_2f | 912 | 82.456 | 82.566 | 82.456 | 82.675 | 80.811 | 78.728 | 78.070 | 77.851 | 84.978 | 79.605 |
| 61 | oocytes_trisopterus_states_5b | 912 | 92.325 | 92.982 | 93.640 | 93.092 | 91.667 | 90.022 | 89.693 | 89.693 | 94.189 | 91.228 |
| 62 | tic-tac-toe | 958 | 99.268 | 99.163 | 99.268 | 99.268 | 99.268 | 99.268 | 99.268 | 99.268 | 98.640 | 100.000 |
| 63 | mammographic | 961 | 72.708 | 71.250 | 72.083 | 75.625 | 77.604 | 78.854 | 78.958 | 79.271 | 80.000 | 78.750 |
| 64 | statlog-german-credit | 1000 | 75.200 | 76.500 | 77.800 | 77.300 | 76.200 | 75.500 | 75.500 | 75.400 | 77.500 | 73.700 |
| 65 | led-display | 1000 | 72.400 | 72.300 | 72.600 | 72.500 | 72.300 | 72.500 | 72.300 | 72.500 | 72.900 | 73.000 |
| 66 | oocytes_merluccius_nucleus_4d | 1022 | 81.078 | 80.588 | 80.686 | 80.686 | 79.412 | 77.255 | 76.961 | 76.765 | 83.725 | 75.490 |
| 67 | oocytes_merluccius_states_2f | 1022 | 92.353 | 91.961 | 92.157 | 92.157 | 91.373 | 90.784 | 90.784 | 90.490 | 93.039 | 92.059 |
| 68 | contrac | 1473 | 40.082 | 44.293 | 47.147 | 50.068 | 51.155 | 52.038 | 52.514 | 51.155 | 50.272 | 43.207 |
| 69 | yeast | 1484 | 42.588 | 49.326 | 54.380 | 58.154 | 60.040 | 60.243 | 60.445 | 60.849 | 59.636 | 54.380 |
| 70 | semeion | 1593 | 93.719 | 93.467 | 93.405 | 93.467 | 93.467 | 93.467 | 93.467 | 93.405 | 96.168 | 95.603 |
| 71 | wine-quality-red | 1599 | 63.062 | 65.938 | 68.812 | 70.000 | 70.625 | 70.375 | 70.312 | 70.312 | 69.625 | 64.438 |
| 72 | plant-texture | 1599 | 83.812 | 81.812 | 79.438 | 77.938 | 77.688 | 77.625 | 77.750 | 77.625 | 86.125 | 85.625 |
| 73 | plant-margin | 1600 | 84.750 | 83.938 | 82.938 | 81.875 | 80.750 | 79.500 | 78.938 | 78.563 | 84.875 | 83.875 |
| 74 | plant-shape | 1600 | 64.812 | 63.562 | 62.438 | 60.375 | 58.312 | 57.062 | 56.375 | 55.937 | 66.250 | 68.000 |
| 75 | car | 1728 | 97.454 | 97.569 | 97.164 | 96.701 | 96.354 | 96.181 | 96.123 | 96.123 | 97.743 | 98.032 |
| 76 | steel-plates | 1941 | 76.289 | 77.320 | 77.938 | 77.423 | 77.062 | 76.753 | 76.598 | 76.495 | 78.351 | 75.103 |
| 77 | cardiotocography-3clases | 2126 | 92.232 | 92.514 | 92.043 | 91.902 | 91.949 | 91.855 | 91.996 | 91.855 | 93.173 | 92.043 |
| 78 | cardiotocography-10clases | 2126 | 80.838 | 82.957 | 82.250 | 80.744 | 79.896 | 79.896 | 79.661 | 79.614 | 84.181 | 79.143 |
| 79 | titanic | 2201 | 78.955 | 78.955 | 78.955 | 78.955 | 78.955 | 78.955 | 78.955 | 78.955 | 78.955 | 78.955 |
| 80 | statlog-image | 2310 | 96.360 | 96.967 | 97.097 | 96.750 | 96.231 | 95.927 | 95.884 | 95.624 | 97.660 | 96.404 |
| 81 | ozone | 2536 | 97.358 | 97.240 | 97.200 | 97.200 | 97.200 | 97.200 | 97.200 | 97.200 | 97.397 | 97.200 |
| 82 | molec-biol-splice | 3190 | 86.731 | 85.947 | 84.536 | 83.093 | 82.465 | 82.403 | 82.371 | 82.371 | 86.920 | 86.418 |
| 83 | chess-krvkp | 3196 | 99.124 | 98.905 | 98.655 | 97.872 | 96.902 | 95.526 | 95.307 | 95.307 | 99.406 | 98.999 |
| 84 | αbalone | 4177 | 50.407 | 49.880 | 55.532 | 60.010 | 62.548 | 64.200 | 64.943 | 65.086 | 63.410 | 64.152 |
| 85 | bank | 4521 | 88.628 | 89.336 | 89.358 | 89.513 | 89.358 | 89.159 | 89.181 | 89.159 | 89.735 | 88.142 |
| 86 | spambase | 4601 | 91.478 | 91.174 | 92.435 | 90.652 | 93.130 | 89.630 | 91.478 | 93.348 | 94.913 | 90.652 |
| 87 | wine-quality-white | 4898 | 63.623 | 66.810 | 67.545 | 68.791 | 69.158 | 68.975 | 68.995 | 68.913 | 69.097 | 65.748 |
| 88 | waveform-noise | 5000 | 86.360 | 86.340 | 86.520 | 86.540 | 86.720 | 86.500 | 85.900 | 85.520 | 86.540 | 85.460 |
| 89 | waveform | 5000 | 85.440 | 85.780 | 86.300 | 86.500 | 86.660 | 86.700 | 86.740 | 86.520 | 86.340 | 84.640 |

Table 4: Comparison between TNTK and MLP-induced NTK for a half of the dataset (3/3).

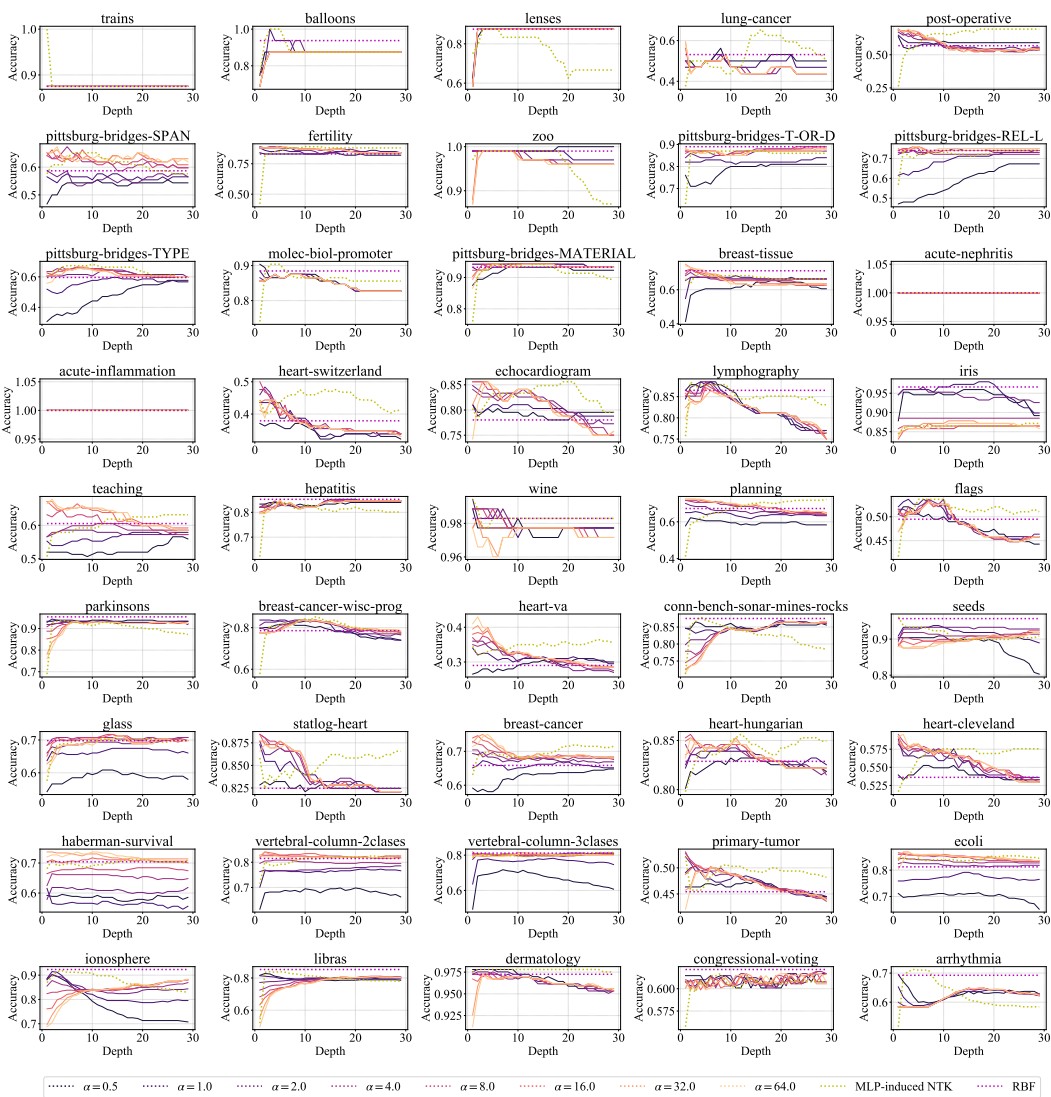

Figure 14: Dataset-wise comparison for a half of the dataset (1/2).

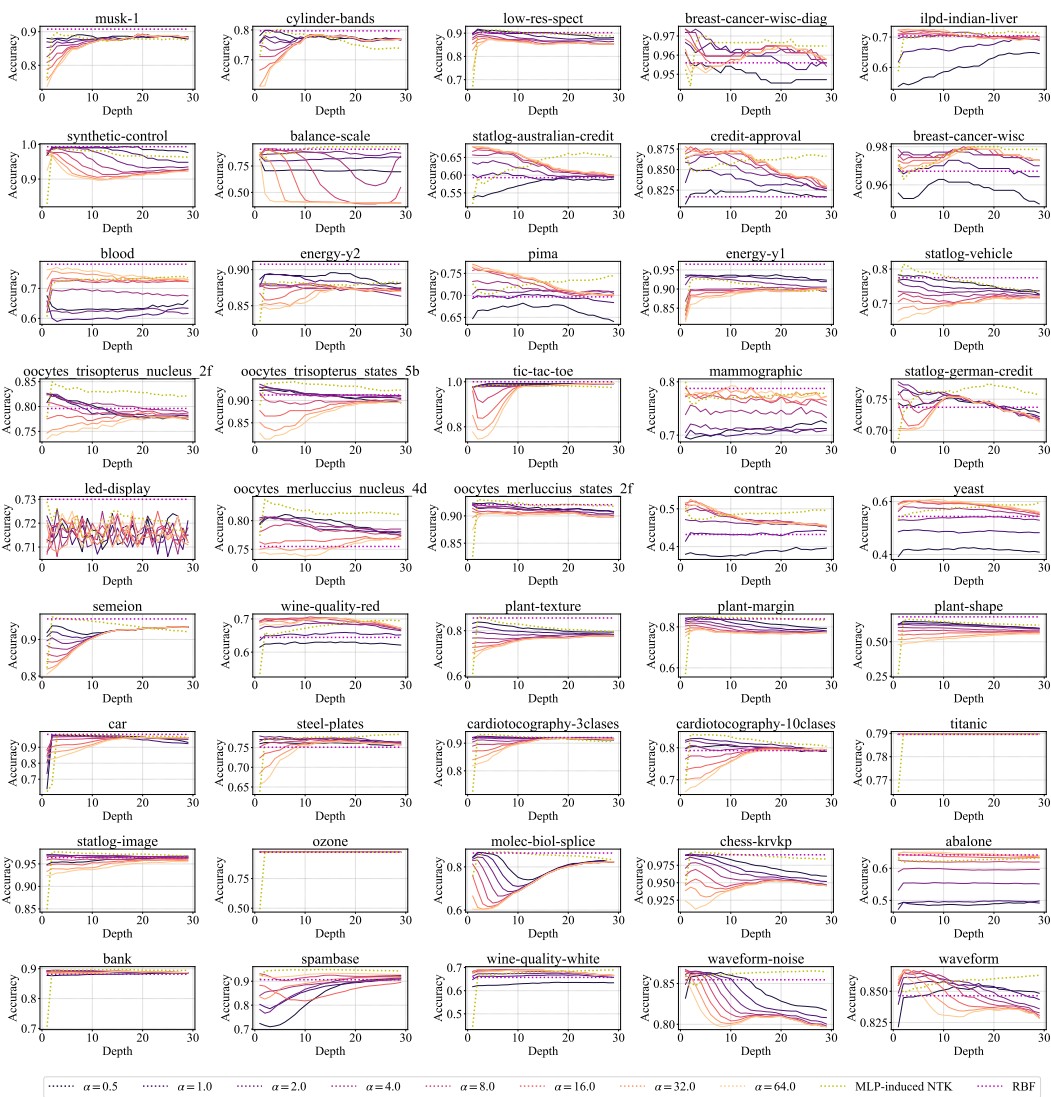

Figure 15: Dataset-wise comparison for a half of the dataset (2/2).

