# OpenReview forum: "A Neural Tangent Kernel Perspective of Infinite Tree Ensembles"
_ICLR.cc/2022/Conference — ICLR 2022 Poster_

### Official Review · Reviewer_3ieY · 2021-10-25

**Correctness:** 4
**Technical Novelty And Significance:** 3
**Empirical Novelty And Significance:** 3
**Recommendation:** 8
**Confidence:** 3

**Main Review:**

The authors define the TNTK and prove a number of properties which one would like to know about the kernel.
Such as positive definiteness, and change during training.
The kernel shares some of the properties of the NTK, indicating that the TNTK can be used to understand training behavior. This is not surprising, considering that soft trees can be viewed as a neural network. This work offers some novelty in computing the kernel in the limit of infinite size of the ensemble, whereas naive application of NTK to large ensembles would be computationally intractible.

The depth degeneracy property is a result with practical utility. The TNTK framework offers a clear understanding of this phenomenon and can help select model hyperparameters.

It is often unclear how to initialize soft-tree ensembles. Many works use classical greedy algorithms to initialize, while others fix the structure of the tree and randomly initialize the weights. It would be nice if the authors can remark whether TNTK can shed some light on initialization strategies.

**Summary Of The Paper:**

The authors derive a neural tangent kernel for soft trees, and prove several properties of the kernel. These include the stability of the kernel in the large ensemble limit, applicability to oblivious tree ensembles, degeneracy with large tree depth, and comparison to NTK. They then evaluate the utility of the kernel on 90 UCI classification data sets using a kernel regression classifier.

**Summary Of The Review:**

This paper represents the development of an important tool for understanding tree ensembles.

---

> ### Author Response · Authors · 2021-11-17
> **Authors’ Response to Reviewer 3ieY**
>
> Thank you for your positive review.
>
> > It would be nice if the authors can remark whether TNTK can shed some light on initialization strategies.
>
> As for the tree structure initialization, since the analysis using the NTK is basically with a fixed model structure, we feel it is not easy to apply the TNTK to analyze strategies such as growing a tree during training.
>
> However, if the strategy is to fix the tree structure and then adjust the parameters, it is possible to take a Neural Architecture Search (NAS)-like approach using the TNTK, as mentioned in Section 5.
> At last year's ICLR, Chen et al. (2021) [1] proposed a method for a Neural Architecture Search (NAS) using the NTK. Their method achieves fast NAS by predicting performance using the kernel without training the model. We believe it is possible to apply this method to the soft tree ensemble to search the tree structure.
> As shown in the right panel of Figure 7, the TNTK computation is more efficient than that of the NTK for typical neural networks. As a result, the TNTK can be useful in the search for a suitable tree structure for a given dataset.
>
> [1] Chen et al. (2021), Neural Architecture Search on ImageNet in Four GPU Hours: A Theoretically Inspired Perspective, ICLR2021

---

### Official Review · Reviewer_yf2B · 2021-11-01

**Correctness:** 4
**Technical Novelty And Significance:** 3
**Empirical Novelty And Significance:** 4
**Recommendation:** 8
**Confidence:** 3

**Main Review:**

This paper proposed a new method called TNTK to ensemble infinite soft trees. It provides non-trivial theoretical studies of the TNTK. Their numerical experiments support their theoretical results. This paper is well-written and clearly organized.

Compared with MLP-induced NTK and RBF, one of the advantages of TNTK will be the low computational complexity. It would be better to provide a detailed complexity analysis (e.g., memory cost, training time complexity, and inference time complexity) of the proposed TNTK.


**Summary Of The Paper:**

This paper proposed the Tree Neural Tangent Kernel (TNTK) for tree ensembles. The proposed idea extends the NTK concept to tree ensemble models and enables ensembles of infinite soft trees. This paper provides theoretical studies to analyze the properties of the proposed TNTK. They also provide comprehensive experimental results to show the effectiveness of the proposed method.

**Summary Of The Review:**

Please refer to my main review.

---

> ### Author Response · Authors · 2021-11-17
> **Authors’ Response to Reviewer yf2B**
>
> Thank you for your positive review.
>
> > It would be better to provide a detailed complexity analysis (e.g., memory cost, training time complexity, and inference time complexity) of the proposed TNTK.
>
> Since the training and inference algorithms of the kernel regression are common across different kernels, we analyze the computational cost of computing a single value in a gram matrix of the corresponding kernels in the following.
>
> The time complexity of the MLP-induced NTK is linear with respect to the layer depth, while that of the TNTK remains to be constant. Such a trend can be seen in the right panel of Figure 7. For the RBF kernel, the computational cost remains the same with respect to changes in hyperparameters, thus its trend is similar to the TNTK.
>
> In terms of the space complexity, when considering a multi-layered MLP, since it is not necessary to store all past calculation results in memory during the recursive computation,  the MLP-induced NTK computation consumes a certain amount of memory regardless of the depth of the layers. Therefore, the memory usage is almost the same across the RBF kernel, TNTK, and MLP-induced NTK.
>
> Since the main text is already 9 pages full, we have included the above complexity analysis in the appendix.

---

### Official Review · Reviewer_r8ZN · 2021-11-02

**Correctness:** 3
**Technical Novelty And Significance:** 2
**Empirical Novelty And Significance:** 2
**Recommendation:** 3
**Confidence:** 2

**Details Of Ethics Concerns:**

A version of the paper has been posted at https://arxiv.org/pdf/2109.04983.pdf  so it is no longer a blind submission.
The conference reaffirms that this is acceptable.


**Main Review:**

The work attempts to generalize the recently proposed Neural Tangent
Kernel method to trees and forests, and attempts to use that to
analyze certain properties and training behavior of a special type of
decision trees (the so-called soft trees) and their infinite ensembles.
This could be of interest to followers of the Neural Tangent
Kernel method and its use in analyzing neural-network-like tree
structures.

For the rest of the audience, the paper has not made an attempt to
clearly lay out all the basics and the logic.  It relies heavily on
external references for many critical concepts and results.

For a start, the notion of a "soft tree" involves some mixing of the
following properties of trees:
(1) whether the splitting function is fixed in one shot (e.g. by
information theoretic choices) or adjustable and trainable (e.g. by
gradient descent) during learning,
(2) during or after training, whether the split is deterministically
applied to every input, and
(3) whether the splitting function parameters are frozen by training or
continue to be samples from some probabilistic distribution that has
parameters estimated during training.
These concepts are never clarified in the paper, so when additional
notions like ensembles and infinitely large ensembles are brought in,
the confusion just grows.

One may wonder how the chosen definition of soft trees is related to
those of probabilistic decision trees [1] and neural trees [2].

[1] Michael I. Jordan,
 A Statistical Approach to Decision Tree Modeling,
 Machine Learning Proceedings 1994, Morgan Kaufmann, 1994, 363-370,
 https://www.sciencedirect.com/science/article/pii/B9781558603356500519

[2] J A Sirat & J-P Nadal,
 Neural trees: A New Tool for Classification,
 Network: Computation in Neural Systems, 1:4, 1990, 423-438,
 https://www.tandfonline.com/doi/abs/10.1088/0954-898X_1_4_003

Given that the work builds on a key concept in prior art, for the
paper to be self-contained, the notion of "Neural Tangent Kernel"
should first be explained (why it is called a Kernel, why Tangent, why
Neural, what is its significance, what new insight it has contributed,
...).   A basic understanding of this is important for the proposed
extension to have a sound footing. It should also be stated upfront
what is expected of the proposed extension.   What insight contributed
by the NTK do you expect to reproduce for tree ensembles?  What new
insight do you want it to bring on?

Many other improvements are needed in the presentation of the logic,
especially on the sudden appearance of certain claims and arguments
around or against them.  For example, when it is declared that trees
of depth larger than 1 have some distinctive feature from MLPs of any
number of layers, what assumption is being argued against?  Did
someone conjecture that they are the same, and why was there such a conjecture?

The paper is not readable due to its lack of clarity in the basic
definitions, a severe lack of connections in the logic, heavy
reliance on external references, and a lack of explanation of the
significance of the results.


**Summary Of The Paper:**

The paper presents an extension of a neural network analytic tool
called Neural Tangent Kernels to its use for decision trees and
forests.  The focus is a previously proposed notion of soft trees.
A few analytical results are presented about the properties of the
extended notion, called Tree Neural Tangent Kernel, such as the
existence of some limiting deterministic form, how alternative
probabilistic tree ensembles would have their kernels converging to
this deterministic form, that this limiting kernel is positive
definite, and that the kernel remains stable during training,
equivalence of the kernels for one-level tree ensembles to those of two-layer perceptrons, etc.
Some support for the analytical claims is provided by simulation experiments.


**Summary Of The Review:**

The paper may have some new results but is better fitting for a forum for theoretical analysis and
preferably a journal where more space is available to fully elaborate on
the detailed definitions, assumptions, claims and their implications.

---

> ### Author Response · Authors · 2021-11-17
> **Authors’ Response to Reviewer r8ZN (1/2)**
>
> Thank you for your detailed comments.
>
> [about the soft tree]
>
> We believe that the points you mentioned have already been written in our original (not revised) submission. We will indicate the relevant sections one by one in the following.
>
> > the notion of a "soft tree" involves some mixing of the following properties of trees…
> > (1) whether the splitting function is fixed in one shot (e.g. by information theoretic choices) or adjustable and trainable (e.g. by gradient descent) during learning
>
> As written in the first line of Section 3, we train model parameters $\boldsymbol{w}$ (internal node parameters) and $\boldsymbol{\pi}$ (leaf parameters). Therefore, the splitting function is adjustable. As you point out, it depends on the settings whether these parameters are updated or fixed, so we have described it in Section 3.
>
> > the notion of a "soft tree" involves some mixing of the following properties of trees…
> > (2) during or after training, whether the split is deterministically applied to every input
>
> Since we are using a scaled error function for $\sigma$ as shown in Equation (5), the deterministic splitting does not happen. The splitting would be deterministic if the output of $\sigma$ takes only 1.0 or 0.0. There may be cases where rounding is included in the function of $\sigma$. However, as you can confirm from Equation (5), our model formulation is not such a case.
>
> > the notion of a "soft tree" involves some mixing of the following properties of trees…
> > (3) whether the splitting function parameters are frozen by training or continue to be samples from some probabilistic distribution that has parameters estimated during training.
>
> In Section 2.1, parameters $\boldsymbol{w}$ (internal node parameters) and $\boldsymbol{\pi}$ (leaf parameters) are defined as real numbers, not probability distributions. Therefore, after training, parameters ($\boldsymbol{w}$ and $\boldsymbol{\pi}$) are fixed (not sampled from any probability distribution). It is the same as the typical training of linear regression and neural networks.
>
> > One may wonder how the chosen definition of soft trees is related to those of probabilistic decision trees [1] and neural trees [2].
>
> As for the probabilistic decision trees proposed in Jordan (1994), their basic idea is to utilize a statistical model for each decision in the decision tree that allows us to utilize ML and MAP estimation techniques. It is not related to our setup.
> As for the neural trees proposed in Sirat & Nadal (1990), since the subject of their paper is about growing binary trees using the tiling algorithm, there is a difference to our problem setup (we fix the tree structure and then adjust the parameters in the model). We have added a note in Section 3 of the revised paper that the tree structure does not change during the gradient descent training.
>
> There are many different formulations of trees. Note that our formulation is based on [1], as shown in the first line of Section 2.1.
>
> [1] Kontschieder et al. (2015), Deep Neural Decision Forests, ICCV2015

---

> > ### Author Response · Authors · 2021-11-17
> > **Authors’ Response to Reviewer r8ZN (2/2)**
> >
> > [about the NTK]
> >
> > We agree that it would be nice to have a more detailed introduction to NTK than we have now, but we argue that our current introduction in the original submission is sufficient enough compared to existing literature. For example, it seems that the amount of introduction is not small compared with the NTK-related papers (e.g., [2][3][4]) that were accepted at ICLR last year.
> >
> > > the notion of "Neural Tangent Kernel" should first be explained (why it is called a Kernel, why Tangent, why Neural, what is its significance, what new insight it has contributed, ...)
> >
> > A brief introduction, including the significance and the contribution of the NTK, is written in Section 2.2 of our original submission. (e.g., in the infinite width limit of the typical neural networks with proper scaling, (1) the training dynamics of gradient flow in function space coincides with kernel ridge-less regression with the limiting NTK, (2) giving a data-dependent generalization bound, (3) training can achieve global convergence if the NTK is positive definite.).
> > In a nutshell, considering the NTK helps in the analytical understanding of the training behavior, as explicitly written right after Equation (4).
> >
> > The reason for the naming of the “Neural Tangent Kernel” is not specifically mentioned in our paper. As can be seen from Equation (3), this is actually the tangent kernel that the neural network models (It does not even have to be neural networks) induce. We believe that the name and the concept of “Neural Tangent Kernel” has been already established and common in the community, for example, [2], [3], and [4] presented at ICLR do not have such explanation.
> >
> > The method for calculating the analytical solution in Figure 4 is also widely known, but in order to make the paper self-contained, we have included the specific formula in the caption of Figure 4 of the revised submission.
> >
> > > What insight contributed by the NTK do you expect to reproduce for tree ensembles? What new insight do you want it to bring on?
> >
> > Thank you for your comment. The goal of this research is to derive the kernel that characterizes the training behavior of soft tree ensembles, and to obtain theoretical support for the empirical techniques such as constraints on individual trees using parameter sharing, adjusting the hardness of the splitting operation, and the use of overparameterization.  We have included the above motivation statement in the introduction of the revised submission.
> > The analysis of the NTK is suitable for the theoretical understanding of the arbitrary (overparameterized) models as described in Section 2.2. We found that such a property is suitable for extensions to tree ensembles, which is our novel contribution in this paper and has not been considered in the past.
> >
> > > when it is declared that trees of depth larger than 1 have some distinctive feature from MLPs of any number of layers, what assumption is being argued against? Did someone conjecture that they are the same, and why was there such a conjecture?
> >
> > Thank you for pointing it out. For NTKs induced by MLPs with depths greater than 2, there was no citation in the relevant section in our original submission. In addition to adding the relevant citation, we have also added a formulation of the MLP-induced NTK that the MLP is deeper than 2 layers in the Appendix of the revised submission.
> >
> > [2] Zhou et al. (2021), Meta-Learning with Neural Tangent Kernel, ICLR2021
> >
> > [3] Alemohammad et al. (2021), The Recurrent Neural Tangent Kernel, ICLR2021
> >
> > [4] Chen and Xu (2021), Deep Neural Tangent Kernel and Laplace Kernel Have the Same RKHS, ICLR2021
> >
> > --------
> >
> > [Others]
> >
> > There are misunderstandings regarding the following two points.
> >
> > > Empirical Novelty And Significance: Not applicable
> >
> > We have numerical experiments in our paper. For example, in Section 5, we have evaluated the classification performance on 90 datasets. At least, we do not think that the assessment of our empirical contribution of the TNTK is "Not applicable".
> >
> > > A version of the paper has been posted at https://arxiv.org/pdf/2109.04983.pdf so it is no longer a blind submission.
> >
> > According to the Call For Papers of ICLR2022, submission of the paper to arXiv is allowed. There is a sentence: "Submission of the paper to archival repositories such as arXiv is allowed" in Dual Submission Policy section.

---

### Decision · Program_Chairs · 2022-01-20

**Decision:**

Accept (Poster)

**Comment:**

The paper investigates the neural tangent kernel NTK of infinitely wide ensembles of soft trees having a particular
soft decision functions in their internal nodes. A closed form of the NTK is presented as well as a result bounding
the changes of the NTK during training. Implications for practical training procedures are briefly discussed and
some experiments are also reported.

The review and discussion phase were difficult with two rather uninformative but positive reviews and a negative
but detailed review. The latter had, however, some serious flaws. For these reasons I carefully read the paper on my own,
too. In turned out that the criticized flaws in the presentation mentioned by the negative reviewer are baseless as long as
one already knows what an NTK is. Given the title of the paper and the history of NTKs, I personally think that
such knowledge can and should be assumed.
Overall, I find the paper be actually very well written. The main issue I see is that the results are not overwhelmingly
surprising. Nonetheless, this is a solid contribution, which deserves to be published.